# RETRIEVAL-BASED CONTROLLABLE MOLECULE GENERATION

**Zichao Wang**[†][*]
Rice University
`jzwang@rice.edu`

**Weili Nie**[†]
NVIDIA
`wnie@nvidia.com`

**Zhuoran Qiao**
Caltech
`zqiao@caltech.edu`

**Chaowei Xiao**
NVIDIA, ASU
`chaoweix@nvidia.com`

**Richard G. Baraniuk**
Rice University
`richb@rice.edu`

**Anima Anandkumar**
NVIDIA, Caltech
`aanandkumar@nvidia.edu`

## ABSTRACT

Generating new molecules with specified chemical and biological properties via generative models has emerged as a promising direction for drug discovery. However, existing methods require extensive training/fine-tuning with a large dataset, often unavailable in real-world generation tasks. In this work, we propose a new retrieval-based framework for controllable molecule generation. We use a small set of exemplar molecules, i.e., those that (partially) satisfy the design criteria, to steer the pre-trained generative model towards synthesizing molecules that satisfy the given design criteria. We design a retrieval mechanism that retrieves and fuses the exemplar molecules with the input molecule, which is trained by a new self-supervised objective that predicts the nearest neighbor of the input molecule. We also propose an iterative refinement process to dynamically update the generated molecules and retrieval database for better generalization. Our approach is agnostic to the choice of generative models and requires no task-specific fine-tuning. On various tasks ranging from simple design criteria to a challenging real-world scenario for designing lead compounds that bind to the SARS-CoV-2 main protease, we demonstrate our approach extrapolates well beyond the retrieval database, and achieves better performance and wider applicability than previous methods.

## 1 INTRODUCTION

Drug discovery is a complex, multi-objective problem (Vamathevan et al., 2019). For a drug to be safe and effective, the molecular entity must interact favorably with the desired target (Parenti and Rastelli, 2012), possess favorable physicochemical properties such as solubility (Meanwell, 2011), and be readily synthesizable (Jiménez-Luna et al., 2021). Compounding the challenge is the massive search space (up to $10^{60}$ molecules Polishchuk et al. (2013)). Previous efforts address this challenge via high-throughput virtual screening (HTVS) techniques (Walters et al., 1998) by searching against existing molecular databases. Combinatorial approaches have also been proposed to enumerate molecules beyond the space of established drug-like molecule datasets. For example, genetic-algorithm (GA) based methods (Sliwoski et al., 2013; Jensen, 2019; Yoshikawa et al., 2018) explore potential new drug candidates via heuristics such as hand-crafted rules and random mutations. Although widely adopted in practice, these methods tend to be inefficient and computationally expensive due to the vast chemical search space (Hoffman et al., 2021). The performance of these combinatorial approaches also heavily depends on the quality of generation rules, which often require task-specific engineering expertise and may limit the diversity of the generated molecules.

To this end, recent research focuses on *learning* to controllably synthesize molecules with *generative models* (Tang et al., 2021; Chen, 2021; Walters and Murcko, 2020). It usually involves first training an unconditional generative model from millions of existing molecules (Winter et al., 2019a; Irwin et al., 2022) and then controlling the generative models to synthesize new desired molecules that

---

[*]Work done during an internship at NVIDIA.
[†]The first two authors contributed equally to this paper.

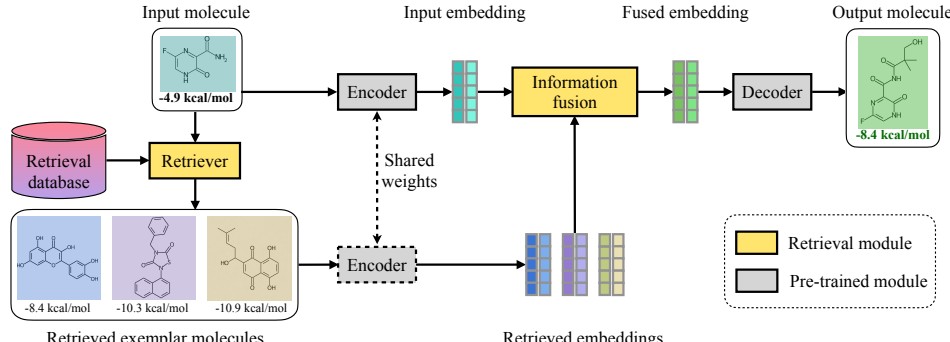

Figure 1: An illustration of RetMol, a retrieval-based framework for controllable molecule generation. The framework incorporates a retrieval module (the molecule retriever and the information fusion) with a pre-trained generative model (the encoder and decoder). The illustration shows an example of optimizing the binding affinity (unit in `kcal/mol`; the lower the better) for an existing potential drug, Favipiravir, for better treating the COVID-19 virus (SARS-CoV-2 main protease, PDB ID: 7L11) under various other design criteria.

satisfy one or more property constraints such as high drug-likeness (Bickerton et al., 2012) and high synthesizability (Ertl and Schuffenhauer, 2009). There are three main classes of learning-based molecule generation approaches: (i) reinforcement-learning (RL)-based methods (Olivecrona et al., 2017; Jin et al., 2020a), (ii) supervised-learning (SL)-based methods (Lim et al., 2018; Shin et al., 2021), and (iii) latent optimization-based methods (Winter et al., 2019b; Das et al., 2021). RL- and SL-based methods train or fine-tune a pre-trained generative model using the desired properties as reward functions (RL) or using molecules with the desired properties as training data (SL). Such methods require heavy task-specific fine-tuning, making them not easily applicable to a wide range of drug discovery tasks. Latent optimization-based methods, in contrast, learn to find latent codes that correspond to the desired molecules, based on property predictors trained on the generative model's latent space. However, training such latent-space property predictors can be challenging, especially in real-world scenarios where we only have a limited number of active molecules for training (Jin et al., 2020b; Huynh et al., 2020). Moreover, such methods usually necessitate a fixed-dimensional latent-variable generative model with a compact and structured latent space (Xue et al., 2018; Xu et al., 2019), making them incompatible with other generative models with a varying-dimensional latent space, such as transformer-based architectures (Irwin et al., 2022).

**Our approach.** In this work, we aim to overcome the aforementioned challenges of existing works and design a controllable molecule generation method that (i) easily generalizes to various generation tasks; (ii) requires minimal training or fine-tuning; and (iii) operates favorably in data-sparse regimes where active molecules are limited. We summarize our contributions as follows:

**[1]** We propose a first-of-its-kind retrieval-based framework, termed RetMol, for controllable molecule generation. It uses a small set of exemplar molecules, which may partially satisfy the desired properties, from a retrieval database to guide generation towards satisfying all the desired properties.

**[2]** We design a retrieval mechanism that retrieves and fuses the exemplar molecules with the input molecule, a new self-supervised training with the molecule similarity as a proxy objective, and an iterative refinement process to dynamically update the generated molecules and retrieval database.

**[3]** We perform extensive evaluation of RetMol on a number of controllable molecule generation tasks ranging from simple molecule property control to challenging real-world drug design for treating the COVID-19 virus, and demonstrate RetMol's superior performance compared to previous methods.

Specifically, as shown in Figure 1, the RetMol framework plugs a lightweight retrieval mechanism into a pre-trained, encoder-decoder generative model. For each task, we first construct a *retrieval database* consisting of exemplar molecules that (partially) satisfy the design criteria. Given an input molecule to be optimized, a *retriever* module uses it to retrieve a small number of exemplar molecules from the database, which are then converted into numerical embeddings, along with the input molecule, by the encoder of the pre-trained generative model. Next, an *information fusion* module fuses the embeddings of exemplar molecules with the input embedding to guide the generation (via the decoder in the pre-trained generative model) towards satisfying the desired properties. The fusion module is the only part in RetMol that requires training. For training, we propose a new

self-supervised objective (*i.e.*, predicting the nearest neighbor of the input molecule) to update the fusion module, which enables RetMol to generalize to various generation tasks without task-specific training/fine-tuning. For inference, we propose a new inference process via iterative refinement that dynamically updates the generated molecules and retrieval database, which leads to improved generation quality and enables RetMol to extrapolate well beyond the retrieval database.

RetMol enjoys several advantages. First, it requires only a handful of exemplar molecules to achieve strong controllable generation performance without task-specific fine-tuning. This makes it particularly appealing in real-world use cases where training or fine-tuning a task-specific model is challenging. Second, RetMol is flexible and easily adaptable. It is compatible with a range of pre-trained generative models, including both fixed-dimensional and varying-dimensional latent-variable models, and requires training only a lightweight, plug-and-play, task-agnostic retrieval module while freezing the base generative model. Once trained, it can be applied to many drug discovery tasks by simply replacing the retrieval database while keeping all other components unchanged.

In a range of molecule controllable generation scenarios and benchmarks with diverse design criteria, our framework achieves state-of-the-art performance compared to latest methods. For example, on a challenging four-property controllable generation task, our framework improves the success rate over the best method by 4.6% (96.9% vs. 92.3%) with better synthesis novelty and diversity. Using a retrieval database of only 23 molecules, we also demonstrate our real-world applicability on a frontier drug discovery task of optimizing the binding affinity of eight existing, weakly-binding drugs for COVID-19 treatment under multiple design criteria. Compared to the best performing approach, our framework succeeds more often at generating new molecules with the given design criteria (62.5% vs. 37.5% success rate) and generates on average more potent optimized drug molecules (2.84 vs. 1.67 `kcal/mol` average binding affinity improvement over the original drug molecules).

## 2 METHODOLOGY: THE RETMOL FRAMEWORK

We now detail the various components in RetMol and how we perform training and inference.

**Problem setup.** We focus on the *multi-property* controllable generation setting in this paper. Concretely, let $x \in \mathcal{X}$ be a molecule where $\mathcal{X}$ denotes the set of all molecules, $a_\ell(x) : \mathcal{X} \to \mathbb{R}$ a property predictor indexed by $\ell \in [1, \dots, L]$, and $\delta_\ell \in \mathbb{R}$ a desired threshold. Then, we formulate multi-property controllable generation as one of three problems below, differing in the control design criteria: (i) a *constraint satisfaction* problem, where we identify a set of new molecules such that $\{x \in \mathcal{X} \mid a_\ell(x) \geq \delta_\ell, \forall \ell\}$, (ii) an *unconstrained optimization* problem, where we find a set of new molecules such that $x' = \operatorname{argmax}_x s(x)$ where $s(x) = \sum_{\ell=1}^{L} w_\ell a_\ell(x)$ with the weighting coefficient $w_\ell$, and (iii) a *constrained optimization* problem that combines the objectives in (i) and (ii).

### 2.1 RETMOL COMPONENTS

**Encoder-decoder generative model backbone.** The pre-trained molecule generative model forms the backbone of RetMol that interfaces between the continuous embedding and raw molecule representations. Specifically, the encoder encodes the incoming molecules into numerical embeddings and the decoder generates new molecules from an embedding, respectively. RetMol is agnostic to the choice of the underlying encoder and decoder architectures, enabling it to work with a variety of generative models and molecule representations. In this work, we consider the SMILES string (Weininger, 1988) representation of molecules and the ChemFormer model (Irwin et al., 2022), which a variant of BART (Lewis et al., 2020) trained on the billion-scale ZINC dataset (Irwin and Shoichet, 2004) and achieves state-of-the-art generation performance.

**Retrieval database.** The retrieval database $\mathcal{X}_R$ contains molecules that can potentially serve as exemplar molecules to steer the generation towards the design criteria and is thus vital for controllable generation. The construction of the retrieval database is task-specific: it usually contains molecules that at least partially satisfy the design criteria in a given task. The domain knowledge of what molecules meet the design criteria and how to select partially satisfied molecules can play an important role in our approach. Thus, our approach is essentially a hybrid system that combines the advantages of both the heuristic-based methods and learning-based methods. Also, we find that a database of only a handful of molecules (e.g., as few as 23) can already provide a strong control signal. This makes our approach easily adapted to various tasks by quickly replacing the retrieval database. Furthermore, the retrieval database can be dynamically updated during inference, i.e., newly generated molecules can enrich the retrieval database for better generalization (see Section 2.3).

**Molecule retriever.** While the entire retrieval database can be used during generation, for computational reasons (e.g., memory and efficiency) it is more feasible to select a small portion of the most relevant exemplar molecules to provide a more accurate guidance. We design a simple heuristic-based retriever that retrieves the exemplar molecules most suitable for the given control design criteria. Specifically, we first construct a "feasible" set containing molecules that satisfy all the given constraints, i.e., $\mathcal{X}' = \cap_{\ell=1}^{L} \{ x \in \mathcal{X}_R \,|\, a_\ell(x) \geq \delta_\ell \}$. If this set is larger than $K$, i.e., the number of exemplar molecules that we wish to retrieve, then we select $K$ molecules with the best property scores, i.e., $\mathcal{X}_r = \text{top}_K(\mathcal{X}', s)$, where $s(x)$ is the task-specific weighted average property score, defined in Section 2. Otherwise, we construct a relaxed feasible set by removing the constraints one at a time until the relaxed feasible set is larger than $K$, at which point we select $K$ molecules with the best scores of the most recently removed property constraints. In either case, the retriever retrieves exemplar molecules with more desirable properties than the input and guides the generation towards the given design criteria. We summarize this procedure in Algorithm 1 in Appendix A. We find that our simple retriever with $K = 10$ works well across a range of tasks. In general, more sophisticated retriever designs are possible and we leave them as the future work.

**Information fusion.** This module enables the retrieved exemplar molecules to modify the input molecule towards the targeted design criteria. It achieves this by merging the embeddings of the input and the retrieved exemplar molecules with a lightweight, trainable, standard cross attention mechanism similar to that in (Borgeaud et al., 2021). Concretely, the fused embedding $e$ is given by

$$e = f_{\text{CA}}(\boldsymbol{e}_{\text{in}}, \boldsymbol{E}_r; \theta) = \text{Attn}(\text{Query}(\boldsymbol{e}_{\text{in}}), \text{Key}(\boldsymbol{E}_r)) \cdot \text{Value}(\boldsymbol{E}_r) \tag{1}$$

where $f_{\text{CA}}$ represents the cross attention function with parameters $\theta$, and $\boldsymbol{e}_{\text{in}}$ and $\boldsymbol{E}_r$ are the input embedding and retrieved exemplar embeddings, respectively. The functions $\text{Attn}$, $\text{Query}$, $\text{Key}$, and $\text{Value}$ compute the cross attention weights and the query, key, and value matrices, respectively. For our choice of the transformer-based generative model (Irwin et al., 2022), we have that $\boldsymbol{e}_{\text{in}} = \text{Enc}(x_{\text{in}}) \in \mathbb{R}^{L \times D}$, $\boldsymbol{E}_r = [\boldsymbol{e}_r^1, \ldots, \boldsymbol{e}_r^K] \in \mathbb{R}^{(\sum_{k=1}^{K} L_k) \times D}$, and $\boldsymbol{e}_r^k = \text{Enc}(x_r^k) \in \mathbb{R}^{L_k \times D}$ where $L$ and $L_k$ are the lengths of the tokenized input and the $k^{\text{th}}$ retrieved exemplar molecules, respectively, and $D$ is the dimension of each token representation. Intuitively, the $\text{Attn}$ function learns to weigh the retrieved exemplar molecules differently such that the more "important" retrieved molecules correspond to the higher weights. The fused embedding $e \in \mathbb{R}^{L \times D}$ thus contains the information of desired properties extracted from the retrieved exemplar molecules, which serves as the input of the decoder to control its generation. More details of this module are available in Appendix A.

## 2.2 TRAINING VIA PREDICTING THE INPUT MOLECULE'S NEAREST NEIGHBOR

The conventional training objective that reconstructs the input molecule (e.g., in ChemFormer (Irwin et al., 2022)) is not appropriate in our case, since perfectly reconstructing the input molecule does not rely on the retrieved exemplar modules (more details in Appendix A). To enable RetMol to learn to use the exemplar molecules for controllable generation, we propose a new self-supervised training scheme, where the objective is to predict the *nearest neighbor* of the input molecule:

$$\mathcal{L}(\theta) = \sum_{i=1}^{\mathcal{B}} \text{CE}\Big( \text{Dec}\big( f_{\text{CA}}(\boldsymbol{e}_{\text{in}}^{(i)}, \boldsymbol{E}_r^{(i)}; \theta) \big), x_{1\text{NN}}^{(i)} \Big). \tag{2}$$

where CE is the cross entropy loss function since we use the BART model as our encoder and decoder (see more details in Appendix A), $x_{1\text{NN}}$ represents the nearest neighbor of the input $x_{\text{in}}$, $\mathcal{B}$ is the batch size, and $i$ indexes the input molecules. The set of retrieved exemplar molecules consists of the remaining $K - 1$ nearest neighbors of the input molecule. During training, we freeze the parameters of the pre-trained encoder and decoder. Instead, we only update the parameters in the information fusion module $f_{\text{CA}}$, which makes our training lightweight and efficient. Furthermore, we use the full training dataset as the retrieval database and retrieve exemplar molecules by best similarity with the input molecule. Thus, our training is not task-specific yet forces the fusion module to be involved in the controllable generation with the similarity as a proxy criterion. We will show that during the inference time, the model trained with the above training objective and proxy criterion using similarity is able to generalize to different generation tasks with different design criteria, and performs better compared to training with the conventional reconstruction objective. For the efficient training, we pre-compute all the molecules' embeddings and their pairwise similarities with efficient approximate $k$NN algorithms (Guo et al., 2020; Johnson et al., 2019).

**Remarks.** The above proposed training strategy that predicts the most similar molecule based on the input molecule and other $K - 1$ similar molecules shares some similarity with masked language

Table 1: Under the similarity constraints, RetMol achieves higher success rate in the constrained QED optimization task and better score improvements in the constrained penalized logP optimization task. Baseline results are reported from (Hoffman et al., 2021).

(a) Success rate of generated molecules that satisfy QED $\in [0.9, 1.0]$ under similarity constraint $\delta = 0.4$.

| Method | Success (%) |
|---|---|
| MMPA (Dalke et al., 2018) | 32.9 |
| JT-VAE (Jin et al., 2018) | 8.8 |
| GCPN (You et al., 2018) | 9.4 |
| VSeq2Seq (Bahdanau et al., 2015) | 58.5 |
| VJTNN+GAN (Jin et al., 2019) | 60.6 |
| AtomG2G (Jin et al., 2019) | 73.6 |
| HierG2G (Jin et al., 2019) | 76.9 |
| DESMILES (Maragakis et al., 2020) | 77.8 |
| QMO (Hoffman et al., 2021) | 92.8 |
| **RetMol** | **94.5** |

(b) The average penalized logP improvements of generated molecules over inputs under similarity constraint $\delta = \{0.6, 0.4\}$.

| Method | Improvement | |
|---|---|---|
| | $\delta = 0.6$ | $\delta = 0.4$ |
| JT-VAE (Jin et al., 2018) | $0.28 \pm 0.79$ | $1.03 \pm 1.39$ |
| GCPN (You et al., 2018) | $0.79 \pm 0.63$ | $2.49 \pm 1.30$ |
| MolDQN (Zhou et al., 2019) | $1.86 \pm 1.21$ | $3.37 \pm 1.62$ |
| VSeq2Seq (Bahdanau et al., 2015) | $2.33 \pm 1.17$ | $3.37 \pm 1.75$ |
| VJTNN (Jin et al., 2019) | $2.33 \pm 1.24$ | $3.55 \pm 1.67$ |
| HierG2G Jin et al. (2019) | $2.49 \pm 1.09$ | $3.98 \pm 1.46$ |
| GA (Nigam et al., 2019) | $3.44 \pm 1.09$ | $5.93 \pm 1.41$ |
| QMO (Hoffman et al., 2021) | $3.73 \pm 2.85$ | $7.71 \pm 5.65$ |
| **RetMol** | $\mathbf{3.78 \pm 3.29}$ | $\mathbf{11.55 \pm 11.27}$ |

model pre-training in NLP (Devlin et al., 2019). However, there are several key differences: 1) we perform "masking" on a sequence level, i.e., a whole molecule, instead of on a token/word level; 2) our "masking" strategy is to predict a particular signal only (i.e., the most similar retrieved molecule to the input) instead of randomly masked tokens; and 3) our training objective is used to only update the lightweight retrieval module instead of updating the whole backbone generative model.

## 2.3 INFERENCE VIA ITERATIVE REFINEMENT

We propose an iterative refinement strategy to obtain an improved outcome when controlling generation with implicit guidance derived from the exemplar molecules. The strategy works by replacing the input $x_{\text{in}}$ and updating the retrieval database with the newly generated molecules. Such an iterative update is common in many other controllable generation approaches such as GA-based methods (Sliwoski et al., 2013; Jensen, 2019) and latent optimization-based methods (Chenthamarakshan et al., 2020; Das et al., 2021). Furthermore, if the retrieval database $\mathcal{X}_R$ is fixed during inference, our method is constrained by the best property values of molecules in the retrieval database, which greatly limits the generalization ability of our approach and also limits our performance for certain generation scenarios such as unconstrained optimization. Thus, to extrapolate beyond the database, we propose to dynamically update the retrieval database over iterations.

We consider the following iterative refinement process. We first randomly perturb the fused embedding $M$ times by adding Guassian noises and greedily decode one molecule from each perturbed embedding to obtain $M$ generated molecules. We score these molecules according to task-specific design criteria. Then, we replace the input molecule with the best molecule in this set if its score is better than that of the input molecule. At the same time, we add the remaining ones to the retrieval database if they have better scores than the lowest score in the retrieval database. If none of the generated molecules has a score better than the input molecule or the lowest in the retrieval database, then the input molecule and the retrieval database stays the same for the next iteration. We also add a stop condition if the maximum allowable number of iterations is achieved or a successful molecule that satisfies the desired criteria is generated. Algorithm 2 in Appendix A summarizes the procedure.

## 3 EXPERIMENTS

We conduct four sets of experiments, covering all controllable generation formulations (see "Problem setup" in Section 2) with increasing difficulty. For fair comparisons, in each experiment we faithfully follow the same setup as the baselines, including using the same number of optimization iterations and evaluating on the same set of input molecules to be optimized. Below, we summarize the main results and defer the detailed experiment setup and additional results to Appendix B and C.

## 3.1 IMPROVING QED AND PENALIZED LOGP UNDER SIMILARITY CONSTRAINT

These experiments aim to generate new molecules that improve upon the input molecule's intrinsic properties including QED (Bickerton et al., 2012) and penalized logP (defined by $\log P(x) - SA(x)$) (Jin et al., 2018) where SA represents synthesizability (Ertl and Schuffenhauer, 2009)) under similarity constraints. For both experiments, we use the top 1k molecules with the best property values from the ZINC250k dataset (Jin et al., 2018) as the retrieval database. In each iteration, we retrieve $K = 20$

exemplar molecules with the best property score that also satisfy the similarity constraint. The remaining configurations exactly follow (Hoffman et al., 2021).

**QED experiment setup and results.** This is a constraint satisfaction problem with the goal of generating new molecules $x'$ such that $a_{\text{sim}}(x', x) \geq \delta = 0.4$ and $a_{\text{QED}}(x') \geq 0.9$ where $x$ is the input molecule, $a_{\text{sim}}$ is the Tanimoto similarity function (Bajusz et al., 2015), and $a_{\text{QED}}$ is the QED predictor. We select 800 molecules with QED in the range $[0.7, 0.8]$ as inputs to be optimized. We measure performance by success rate, i.e., the percentage of input molecules that result in a generated molecule that satisfy both constraints. Table 1a shows the success rate of QED optimization by comparing RetMol with various baselines. We can see that RetMol achieves the best success rate, e.g., 94.5% versus 92.8% compared to the best existing approach.

**Penalized logP setup and results.** This is a constrained optimization problem with the goal to generate new molecules $x'$ to maximize the penalized logP value $a_{\text{plogP}}(x')$ with similarity constraint $a_{\text{sim}}(x', x) \geq \delta$ where $\delta \in \{0.4, 0.6\}$. We select 800 molecules that have the lowest penalized logP values in the ZINC250k dataset as inputs to be optimized. We measure performance by the relative improvement in penalized logP between the generated and input molecules, averaged over all 800 input molecules. Table 1b shows the results comparing RetMol to various baselines. RetMol outperforms the best existing method for both similarity constraint thresholds and, for the $\delta = 0.4$ case, improves upon the best existing method by almost 50%. The large variance in RetMol is due to large penalized logP values of a few generated molecules, and is thus not indicative of poor statistical significance. We provide a detailed analysis of this phenomenon in Appendix C.

## 3.2 OPTIMIZING GSK3$\beta$ AND JNK3 INHIBITION UNDER QED AND SA CONSTRAINTS

This experiment aims to generate novel, strong inhibitors for jointly inhibiting both GSK3$\beta$ (Glycogen synthase kinase 3 beta) and JNK3 (-Jun N-terminal kinase-3) enzymes, which are relevant for potential treatment of Alzheimer's disease (Jin et al., 2020a; Li et al., 2018). Following (Jin et al., 2020a), we formulate this task as a constraint satisfaction problem with four property constraints: two positivity constraints $a_{\text{GSK3}\beta}(x') \geq 0.5$ and $a_{\text{JNK3}}(x') \geq 0.5$, and two molecule property constraints of QED and SA that $a_{\text{QED}}(x') \geq$

Table 2: Success rate, novelty and diversity of generated molecules in the task of optimizing four properties: QED, SA, and two binding affinities to GSK3$\beta$ and JNK3 estimated by pre-trained models from (Jin et al., 2020a). Baseline results are reported from Jin et al. (2020a); Xie et al. (2021).

| Method | Success % | Novelty | Diversity |
|---|---|---|---|
| JT-VAE (Jin et al., 2018) | 1.3 | - | - |
| GVAE-RL (Jin et al., 2020a) | 2.1 | - | - |
| GCPN (You et al., 2018) | 4.0 | - | - |
| REINVENT (Olivecrona et al., 2017) | 47.9 | 0.561 | 0.621 |
| RationaleRL (Jin et al., 2020a) | 74.8 | 0.568 | 0.701 |
| MARS (Xie et al., 2021) | 92.3 | 0.824 | 0.719 |
| MolEvol (Chen* et al., 2021) | 93.0 | 0.757 | 0.681 |
| **RetMol** | **96.9** | **0.862** | **0.732** |

0.6 and $a_{\text{SA}}(x') \leq 4$. Note that $a_{\text{GSK3}\beta}$ and $a_{\text{JNK3}}$ are property predictors (Jin et al., 2020a; Li et al., 2018) for GSK3$\beta$ and JNK3, respectively, and higher values indicate better inhibition against the respective proteins. The retrieval database consists of all the molecules from the CheMBL (Gaulton et al., 2016) dataset (approx. 700) that satisfy the above four constraints and we retrieve $K = 10$ exemplar molecules most similar to the input each time. For evaluation, we compute three metrics including success rate, novelty, and diversity according to (Jin et al., 2020a).

Table 2 shows that RetMol outperforms all previous methods on the four-property molecule optimization task. In particular, our framework achieves these results without task-specific fine-tuning required for RationaleRL and REINVENT. Besides, RetMol is computationally much more efficient than MARS, which requires 550 iterations model training whereas RetMol requires only 80 iterations (see more details in Appendix B.5). MARS also requires test-time, adaptive model training per sampling iteration which further increases the computational overhead during generation.

## 3.3 GUACAMOL BENCHMARK MULTIPLE PROPERTY OPTIMIZATION

This experiment evaluates RetMol on the seven multiple property optimization (MPO) tasks, a subset of tasks in the Guacamol benchmark (Brown et al., 2019). They are the unconstrained optimization problems to maximize a weighted sum of multiple diverse molecule properties. We choose these MPO tasks because 1) they represent a more challenging subset in the benchmark, as existing methods can achieve almost perfect performance on most of the other tasks (Brown et al., 2019), and 2) they are the most relevant tasks to our work, e.g., multi-property optimization. The retrieval database

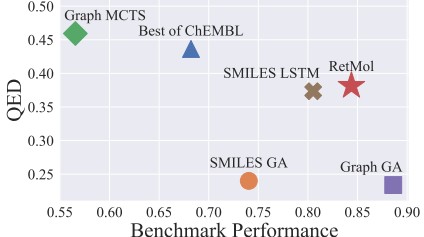 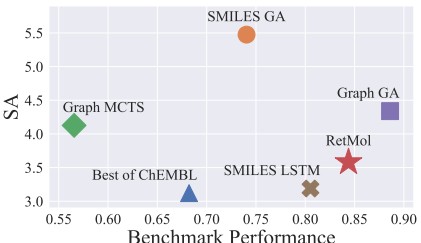

Figure 2: Comparison with the state-of-the-art methods in the multiple property optimization (MPO) tasks on the Guacamol benchmark. **Left**: QED (↑) versus the averaged benchmark performance (↑). **Right**: SA (↓) versus the average benchmark performance (↑). RetMol achieves the best balance between improving the benchmark performance while maintaining the synthesizability (SA) and drug-likeness (QED) of generated molecules.

Table 3: Quantitative results in the COVID-19 drug optimization task, where we aim to improve selected molecules' binding affinity (estimated via docking (Shidi et al., 2022)) to the SARS-CoV-2 main protease under the QED, SA, and similarity constraints. Under stricter similarity condition, RetMol succeeds in more cases (5/8 versus 3/8). Under milder similarity condition, RetMol achieves higher improvements (2.84 versus 1.67 average binding affinity improvements). Unit of numbers in the table is `kcal/mol` and lower is better.

| Input molecule | Input score | $\delta = 0.6$ | | $\delta = 0.4$ | |
| | | RetMol | Graph GA (Jensen, 2019) | RetMol | Graph GA (Jensen, 2019) |
|---|---|---|---|---|---|
| Favipiravir | -4.93 | -6.48 | **-7.10** | **-8.70** | -7.10 |
| Bromhexine | -9.64 | **-11.48** | -11.20 | **-12.65** | -11.83 |
| PX-12 | -6.13 | **-8.45** | -8.07 | **-10.90** | -8.31 |
| Ebselen | -7.31 | - | - | **-10.82** | -10.41 |
| Disulfiram | -8.58 | **-9.09** | - | **-10.44** | -10.00 |
| Entecavir | -9.00 | - | - | **-12.34** | - |
| Quercetin | -9.25 | - | - | **-9.84** | 9.81 |
| Kaempferol | -8.45 | **-8.54** | - | **-10.35** | 10.19 |
| **Avg. Improvement** | - | **0.78** | 0.71 | **2.84** | 1.67 |

consists of 1k molecules with best scores for each task and we retrieve $K = 10$ exemplar molecules with the highest scores each time.

We demonstrate that RetMol achieves the best results along the Pareto frontier of the molecular design space. Figure 2 visualizes the benchmark performance averaged over the seven MPO tasks against QED, SA, two metrics that evaluate the drug-likeness and synthesizability of the optimized molecules and that are not part of the Guacamol benchmark's optimization objective. Our framework achieves a nice balance between optimizing benchmark performance and maintaining good QED and SA scores. In contrast, Graph GA (Jensen, 2019) achieves the best benchmark performance but suffers from low QED and high SA scores. These results demonstrate the advantage of retrieval-based controllable generation: because the retrieval database usually contains drug-like molecules with desirable properties as high QED and low SA, the generation is guided by these molecules in a good way to not deviate too much from these desirable properties. Moreover, because the retrieval database is updated with newly generated molecules with better benchmark performance, the generation can produce molecules with benchmark score beyond the best in the initial retrieval database. Additional results in Figure 2 in Appendix C.3 corroborate with those presented above.

### 3.4 OPTIMIZING EXISTING INHIBITORS FOR SARS-CoV-2 MAIN PROTEASE

To demonstrate our framework's applicability at the frontiers of drug discovery, we apply it to the real-world task of improving the inhibition of existing weak inhibitors against the SARS-CoV-2 main protease (M$^{pro}$, PDB ID: 7L11), which is a promising target for treating COVID-19 by neutralizing the SARS-CoV-2 virus (Gao et al., 2022; Zhang et al., 2021). Because the the novel nature of this virus, there exist few high-potency inhibitors, making it challenging for learning-based methods that require a sizable training set not yet attainable in this case. However, the few existing inhibitors make excellent candidates for the retrieval database in our framework. We use a set of 23 known inhibitors (Jin et al., 2020b; Huynh et al., 2020; Hoffman et al., 2021) as the retrieval database and select the 8 weakest inhibitors to M$^{pro}$ as input. We design an optimization task to maximize the binding affinity (estimated via docking (Shidi et al., 2022); see Appendix B for the detailed procedure) between the generated molecule and M$^{pro}$ while satisfying the following three constraints: $a_{\text{QED}}(x') \geq 0.6$, $a_{\text{SA}}(x') \leq 4$, and $a_{\text{sim}}(x', x) \geq \delta$ where $\delta \in \{0.4, 0.6\}$.

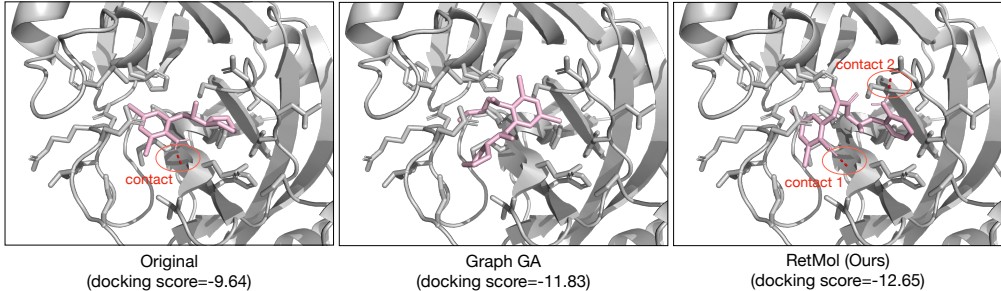



Original (docking score=-9.64)     Graph GA (docking score=-11.83)     RetMol (Ours) (docking score=-12.65)



Figure 3: 3D visualizations that compare RetMol with Graph GA in optimizing the original inhibitor, Bromhexine, that binds to the SARS-CoV-2 main protease in the $\delta = 0.6$ case. We can see the optimized inhibitor in RetMol has more polar contacts (red dotted lines) and also more disparate binding modes with the original compound than the Graph GA optimized inhibitor, which aligns with the quantitative results.

Table 4: **Left**: Comparing different training schemes in the unconditional generation setting. **Right**: Generation performance with different retrieval database constructions based on the experiment in Section 3.2.

| Training objective | Validity | Novelty | Uniqueness |
|---|---|---|---|
| RetMol (predict NN) | **0.902** | **0.998** | **0.922** |
| Conventional (recon. input) | 0.834 | 0.998 | 0.665 |

| Ret. database construction | Success % | Novelty | Diversity |
|---|---|---|---|
| GSK3$\beta$ + JNK3 + QED + SA | **96.9** | **0.862** | **0.732** |
| GSK3$\beta$ + JNK3 | 84.7 | 0.736 | 0.700 |
| GSK3$\beta$ or JNK3 | 44.1 | 0.571 | 0.708 |

Table 3 shows the optimization results of comparing RetMol with graph GA, a competitive baseline. Under stricter ($\delta = 0.6$) similarity constraint, RetMol successfully optimizes the most molecules constraint (5 out of 8) while graph GA fails to optimize input molecules most of the time because it cannot satisfy the constraints or improve upon the input drugs' binding affinity. Under milder ($\delta = 0.4$) similarity constraint, our framework achieves on average higher improvement (2.84 versus 1.67 binding affinity improvements) compared to the baseline. We have also tested QMO (Hoffman et al., 2021), and find it unable to generate molecules that satisfy all the given constraints.

We produce 3D visualizations in Figure 3 (along with a video demo in `https://shorturl.at/fmtyS`) to show the comparison of the RetMol-optimized inhibitors that bind to the SARS-CoV-2 main protease with the original and GA-optimized inhibitors. We can see that 1) there are more polar contacts (shown as red dotted lines around the molecule) in the RetMol-optimized compound, and 2) the binding mode of the GA-optimized molecule is much more similar to the original compound than the RetMol-optimized binding mode, implying RetMol optimizes the compound beyond local edits to the scaffold. These qualitative and quantitative results together demonstrate that RetMol has the potential to effectively optimize inhibitors in a real-world scenario. Besides, we provide extensive 2D graph visualizations of optimized molecules in Tables A3 and A4 in the Appendix. Finally, we also perform another set of experiments on Antibacterial drug design for the MurD protein (Sangshetti et al., 2017) and observe similar results; see Appendix C.5 for more details.

## 3.5 ANALYSES

We analyze how the design choices in our framework impact generation performance. We summarize the main results and defer the detailed settings and additional results to Appendix B and C. Unless otherwise stated, we perform all analyses on the experiment setting in Section 3.2.

**Training objectives.** We evaluate how the proposed self-training objective in Sec. 2.2 affects the quality of generated molecules in the unconditional generation setting. Table 4 (**left**) shows that, training with our proposed nearest neighbor objective (i.e., predict NN) indeed achieves better generation performance than the conventional input reconstruction objective (i.e., recon. input).

**Types of retrieval database.** We evaluate how the retrieval database construction impacts controllable generation performance, by comparing four different constructions: with molecules that satisfy all four constraints, i.e., GSK3$\beta$ + JNK3 + QED + SA; or satisfy only GSK3$\beta$ + JNK3; or satisfy only GSK3$\beta$ or JNK3 but not both. Table 4 (**right**) shows that a retrieval database that better satisfies the design criteria and better aligns with the controllable generation task generally leads to better performance. Nevertheless, we note that RetMol performs reasonably well even with exemplar molecules that only partially satisfy the properties (e.g., GSK3$\beta$ + JNK3), achieving comparable performance to RationaleRL (Jin et al., 2020a).

**Size of retrieval database.** Figure 4 (**left**) shows a larger retrieval database generally improves all metrics and reduces the variance. It is particularly interesting that our framework achieves strong

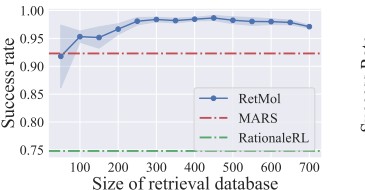 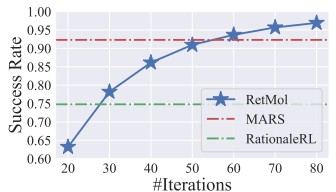 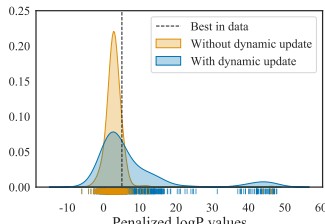

Figure 4: Generation performance with varying retrieval database size (**left**), varying number of iterations (**middle**), and with or without dynamically updating the retrieval database (**right**). The left two plots are based on the experiment in Section 3.2 while the right plot is based on the penalized logP experiment in Section 3.1.

performance with a small retrieval database, which already outperforms the best baseline on the success rate metric with only a 100-molecule retrieval database.

**Number of optimization iterations.** Figure 4 (**middle**) shows that RetMol outperforms the best existing methods with a small number of iterations (outperforms RationaleRL at 30 iterations and MARS at 60 iterations); its performance continues to improve with increasing iterations.

**Dynamic retrieval database update.** Figure 4 (**right**) shows that, for the penalized logP optimization experiment (Section 3.1), with dynamical update our framework generates more molecules with property values beyond the best in the data compared to the case without dynamic update. This comparison shows that dynamic update is crucial to generalizing beyond the retrieval database.

## 4 RELATED WORK

RetMol is most related to controllable molecule generation methods, which we have briefly reviewed and compared with in Section 1. Another line of work in this direction leverages constraints based on *explicit*, pre-defined molecule scaffold structures to guide the generative process (Maziarz et al., 2022; Langevin et al., 2020). Our approach is fundamentally different in that the retrieved exemplar molecules *implicitly* guides the generative process through the information fusion module.

RetMol is also inspired by a recent line of work that integrates a retrieval module in various NLP and vision tasks, such as language modeling (Borgeaud et al., 2021; Liu et al., 2022; Wu et al., 2022), code generation (Hayati et al., 2018), question answering (Guu et al., 2020; Zhang et al., 2022), and image generation (Tseng et al., 2020; Casanova et al., 2021; Blattmann et al., 2022; Chen et al., 2022). Among them, the retrieval mechanism is mainly used for improving the generation quality either with a smaller model (Borgeaud et al., 2021; Blattmann et al., 2022) or with very few data points (Casanova et al., 2021; Chen et al., 2022). None of them has explicitly explored the *controllable* generation with retrieval. The retrieval module also appears in bio-related applications such as multiple sequence alignment (MSA), which can be seen as a way to search and retrieve relevant protein sequences, and has been an essential building block in MSA transformer (Rao et al., 2021) and AlphaFold (Jumper et al., 2021). However, the MSA methods focus on the pairwise interactions among a set of evolutionarily related (protein) sequences while RetMol considers the cross attention between the input and a set of retrieved examples.

There also exist priors work that study retrieval-based approaches for controllable text generation (Kim et al., 2020; Xu et al., 2020). These methods require task-specific training/fine-tuning of the generative model for each controllable generation task whereas our framework can be applied to many different tasks without it. While we focus on molecules, our framework is general and has the potential to achieve controllable generation for multiple other modalities beyond molecules.

## 5 CONCLUSIONS

We proposed RetMol, a new retrieval-based framework for controllable molecule generation. By incorporating a retrieval module with a pre-trained generative model, RetMol leverages exemplar molecules retrieved from a task-specific retrieval database to steer the generative model towards generating new molecules with the desired design criteria. RetMol is versatile, requires no task-specific fine-tuning and is agnostic to the generative models (see Appendix C.7). We demonstrated the effectiveness of RetMol on a variety of benchmark tasks and a real-world inhibitor design task for the SARS-CoV-2 virus, achieving state-of-the-art performances in each case comparing to existing methods. Since RetMol still requires exemplar molecules that at least partially satisfy the design criteria, it becomes challenging when those molecules are unavailable at all. A valuable future work is to improve the retrieval mechanisms such that even weaker molecules, i.e., those that do not satisfy but are close to satisfying the design criteria, can be leveraged to guide the generation process.

## ACKNOWLEDGEMENTS

ZW and RGB are supported by by NSF grants CCF-1911094, IIS-1838177, and IIS-1730574; ONR grants N00014-18-12571, N00014-20-1-2534, and MURI N00014-20-1-2787; AFOSR grant FA9550-22-1-0060; and a Vannevar Bush Faculty Fellowship, ONR grant N00014-18-1-2047.

## ETHICS STATEMENT

Applications that involve molecule generation such as drug discovery are high-stake in nature. These applications are highly regulated to prevent potential misuse (Hill and Richards, 2022). RetMol as a technology to improve controllable molecule generation has the potential to be subjected to malicious use. For example, one could change the retrieval database and the design criteria into harmful ones, such as increased drug toxicity. However, we note that RetMol is a computational tool useful for *in silico* experiments. As a result, although RetMol can suggest new molecules according to arbitrary design criteria, the properties of the generated molecules are estimations of the real chemical and biological properties and need to be further validated in lab experiments. Thus, while RetMol's real-world impact is limited to *in silico* experiments, it is also prevented from directly generating real drugs that can be readily used. In addition, controllable molecule generation is an active area of research; we hope that our work contribute to this ongoing line of research and make ML methods safe and reliable for molecule generation applications in the real world.

## REPRODUCIBILITY STATEMENT

To ensure the reproducibility of the empirical results, we provide the implementation details of each task (*i.e.*, experimental setups, hyperparameters, dataset specifications, etc.) in Appendix B. The source code is available at `https://github.com/NVlabs/RetMol`.

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

# Appendix

## A  FRAMEWORK DETAILS

**More details of the information fusion module (Eq. ( 1))**  Recall that Eq.( 1) is

$$\boldsymbol{e} = f_{\mathrm{CA}}(\boldsymbol{e}_{\mathrm{in}}, \boldsymbol{E}_r; \theta) = \mathrm{Attn}(\mathrm{Query}(\boldsymbol{e}_{\mathrm{in}}), \mathrm{Key}(\boldsymbol{E}_r)) \cdot \mathrm{Value}(\boldsymbol{E}_r)$$

where $f_{\mathrm{CA}}$ represents the cross attention function with parameters $\theta$, and $\boldsymbol{e}_{\mathrm{in}} \in \mathbb{R}^{L \times D}$ and $\boldsymbol{E}_r \in \mathbb{R}^{(\sum_{k=1}^{K} L_k) \times D}$ are the input embedding and retrieved exemplar embeddings, respectively.

Concretely, $\mathrm{Query}$, $\mathrm{Key}$, and $\mathrm{Value}$ are affine functions that do not change the shape of the the input. The function input and output dimensions are as follows:

$$\mathrm{Query} : \mathbb{R}^{L \times D} \to \mathbb{R}^{L \times D} , \tag{3}$$

$$\mathrm{Key} : \mathbb{R}^{(\sum_{k=1}^{K} L_k) \times D} \to \mathbb{R}^{(\sum_{k=1}^{K} L_k) \times D} , \tag{4}$$

$$\mathrm{Value} : \mathbb{R}^{(\sum_{k=1}^{K} L_k) \times D} \to \mathbb{R}^{(\sum_{k=1}^{K} L_k) \times D} \tag{5}$$

In particular, the $\mathrm{Key}$ and $\mathrm{Value}$ functions first apply to each of the $k$-th retrieved molecule embedding $\boldsymbol{e}_r^k \in \mathbb{R}^{L_k \times D}$ (which is a block of matrix in the retrieved molecule embedding $\boldsymbol{E}_r$) and then stack these K output matrices horizontally to obtain the output matrices of shape $(\sum_{k=1}^{K} L_k) \times D$.

The $\mathrm{Attn}$ function first computes the inner product between each slice of the query output matrix (of shape $D$) and the key output matrix (of shape $(\sum_{k=1}^{K} L_k) \times D$), followed by a softmax function which results in $(\sum_{k=1}^{K} L_k)$ un-normalized weights. These weights are applied to the value output matrix (of shape $(\sum_{k=1}^{K} L_k) \times D$) to obtain one slice in the output fused matrix $\boldsymbol{e}$. The above procedure is performed in parallel to obtain the full output fused matrix $\boldsymbol{e}$, such that it maintains the same dimensionality with the input embedding $\boldsymbol{e}_{\mathrm{in}}$, i.e., $\boldsymbol{e} \in \mathbb{R}^{L \times D}$.

**More details of the CE loss**  The cross entropy in Eq. (2) is the maximum likelihood objective commonly used in sequence modeling and sequence-to-sequence generation tasks such as machine translation. It takes two molecules as its inputs: one is the ground-truth molecule sequence, and the other one is the generated molecule sequence (i.e., the softmax output of decoder). Specifically, if we define $y := x_{1\mathrm{NN}}$ and $\hat{y} := \mathrm{Dec}(f_{\mathrm{CA}}(e_{\mathrm{in}}, E_r); \theta)$, then the "CE" in Eq. (2) is given as follows:

$$\mathrm{CE}(\hat{y}, y) = - \sum_{l=0}^{L-1} \sum_{v=0}^{V-1} y_{l,v} \log \hat{y}_{l,v}$$

where $L$ is the molecule sequence length, $V$ is the vocabulary size, $y_{l,v}$ is the ground-truth (one-hot) probability of vocabulary entry $v$ on the $l$-th token, and $\hat{y}_{l,v}$ is the predicted probability (i.e., softmax output of decoder) of the vocabulary entry $v$ on the $l$-th token.

**More details of the RetMol training objective (Sec. 2.2)**  We first provide more detailed descriptions of our proposed objective: Given an input molecule, we use its nearest neighbor molecule from the retrieval database, based on the cosine similarity in the embedding space of the CDDD encoder, as the prediction target. The decoder takes in the fused embeddings from the information fusion module to generate new molecules. As shown in Eq. (2), we calculate the cross-entropy distance between the decoded output and the nearest neighbor molecule as the training objective.

The motivation is that: Other similar molecules (i.e., the remaining $K - 1$ nearest neighbors) from the retrieval database, through the fusion module, can provide good guidance for transforming the input to its nearest neighbor (e.g., how to perturb the molecule and how large the perturbation would be). Accordingly, the fusion module can be effectively updated through this auxiliary task.

On the contrary, if we use the conventional encoder-decoder training objective (i.e., reconstructing the input molecule), the method actually does not need anything from the retrieval database to do well in the input reconstruction (as the input molecule itself contains all the required information already). As a result, the information fusion module would not be effectively updated during training.

**Molecule retriever** Algorithm 1 describes the how the retriever retrieves exemplar molecules from a retrieval database given the design criteria.

**Inference** Algorithm 2 describes the inference procedure.

**Parameters** The total number of parameters in RetMol and in the base generative model (Irwin et al., 2022) is 10471179 and 10010635, respectively. The additional 460544 parameters come exclusively from the information fusion model, which means that it adds only less than 5% parameter overhead and is very lightweight compared to the base generative model.

---

**Algorithm 1:** Exemplar molecule retriever

---

**Require :** Property predictors $a_\ell$ and desired property thresholds $\delta_\ell$ for $\ell \in [1, \dots, L]$ for
     property constraints; scoring function $s$ for properties to be optimized; retrieval
     database $\mathcal{X}_R$; number of retrieved exemplar molecules $K$

**Input**  :Input molecule $x_{\text{in}}$

**Output** :Retrieved exemplar molecules $\mathcal{X}_r$

1 $\ell' = L$;

2 $\mathcal{X}' = \cap_{\ell=1}^{L} \{x \in \mathcal{X}_R \,|\, a_\ell(x) \geq \delta_\ell\}$;

3 **while** $|\mathcal{X}'| \leq K$ **do**

4  │ $L := L - 1$;

5  │ $\ell' := L$;

6  │ $\mathcal{X}' := \cap_{\ell=1}^{L} \{x \in \mathcal{X}_R \,|\, a_\ell(x) \geq \delta_\ell\}$;

7 **end**

8 $\mathcal{X}'' = \{x \in \mathcal{X}' \,|\, a_{\ell'}(x) \geq a_{\ell'}(x_{\text{in}})\}$;

9 **if** $|\mathcal{X}''| \geq K$ **then**

10 │ **return** $\mathcal{X}_r = \text{top}_K(\mathcal{X}'', s)$;

11 **else**

12 │ **return** $\mathcal{X}_r = \text{top}_K(\mathcal{X}', s)$;

13 **end**

---

## B DETAILED EXPERIMENT SETUP

### B.1 RETMOL TRAINING

We only train the information fusion model in our RetMol framework. The training dataset uses either ZINC250k (Irwin and Shoichet, 2004) (for the experiments in Section 3.1 or CheMBL (Gaulton et al., 2016). The choice of the training data is to align with the setup for the baseline methods in each experiment; The experiments in Sections 3.1 and 3.3 use the ZINC250k dataset while the experiments in Sections 3.2 and 3.4 use the CheMBL dataset. For the ZINC250k dataset, we follow the train/validation/test splits in (Jin et al., 2019) and train on the train split. For the ChemMBL dataset, we train on the entire dataset without splits. Training is distributed over four V100 NVIDIA GPUs, each with 16GB memory, with a batch size of 256 samples on each GPU, for 50k iterations. The total training time is approximately 2 hours for either the ZINC250k or the CheMBL dataset. Our training infrastructure largely follows the Megatron[1] version of the molecule generative model in (Irwin et al., 2022), which uses DeepSpeed,[2] Apex,[3] and half precision for improved training efficiency. Note that, once RetMol is trained, we do not perform further task-specific fine-tuning.

---

[1]https://github.com/NVIDIA/Megatron-LM

[2]https://github.com/microsoft/DeepSpeed

[3]https://github.com/NVIDIA/apex

---

**Algorithm 2:** Generation with adaptive input and retrieval database update

---

**Require:** Encoder Enc, decoder Dec, information fusion model $f_{\mathrm{CA}}$, exemplar molecule
retriever Ret, property predictors $a_\ell(x)$ and desired property thresholds $\delta_\ell$ for
$\ell \in [1, \ldots, L]$ for property constraints, retrieval database $\mathcal{X}_R$, scoring function $s(x)$
for properties to be optimized

**Input** : Input molecule $x_{\mathrm{in}}$, number of retrieved exemplar molecules $K$, number of
optimization iterations $T$, the number of molecules $M$ to sample at each iteration

**Output** : Optimized molecule $x'$

1 **for** $t \in [1, \ldots, T]$ **do**
2 $\quad$ $\boldsymbol{e}_{\mathrm{in}} = \mathrm{Enc}(x)$;
3 $\quad$ $\mathcal{X}_r = \mathrm{Ret}\big(\mathcal{X}_R, \{a_\ell, \delta_\ell\}_{\ell=1}^L, s, x_{\mathrm{in}}, K\big)$;
4 $\quad$ $\boldsymbol{E}_r = \mathrm{Enc}(\mathcal{X}_r)$;
5 $\quad$ $\boldsymbol{e} = f_{\mathrm{CA}}(\boldsymbol{e}_{\mathrm{in}}, \boldsymbol{E}_r; \theta)$;
6 $\quad$ $\mathcal{X}_{\mathrm{gen}} = \emptyset$;
7 $\quad$ **for** $i = 1, \ldots, M$ **do**
8 $\quad\quad$ $\epsilon \sim \mathcal{N}(0, \mathbf{1})$;
9 $\quad\quad$ $\boldsymbol{e}_i = \boldsymbol{e} + \epsilon$;
10 $\quad\quad$ $x' = \mathrm{Dec}(\boldsymbol{e}_i)$;
11 $\quad\quad$ $\mathcal{X}_{\mathrm{gen}} := \mathcal{X}_{\mathrm{gen}} \cup \{x'\}$;
12 $\quad$ **end**
13 $\quad$ $\mathcal{X}_{\mathrm{gen}} := \mathrm{Ret}\big(\mathcal{X}_{\mathrm{gen}}, \{a_\ell, \delta_\ell\}_{\ell=1}^L, s, x_{\mathrm{in}}, K\big)$;
14 $\quad$ $x_{\mathrm{in}} := \mathrm{top}_1(\mathcal{X}_{\mathrm{gen}}, s)$;
15 $\quad$ $\mathcal{X}_R := \mathcal{X}_R \cup \mathcal{X}_{\mathrm{gen}} \setminus x_{\mathrm{in}}$
16 **end**

### B.2 RETMOL INFERENCE

We use greedy sampling in the decoder throughout all experiments in this paper. When we need to sample more than one molecule from the decoder given the (fused) embedding, we first perturb the (fused) embedding with independent random isotropic Gaussian with standard deviation of 1 and then generate a sample from each perturbed (fused) embedding using the decoder. Inference uses a single V100 NVIDIA GPU with 16 GB memory.

Note that during inference, the multi-property design objective is provided with a general form $s(x) = \sum_{l=1}^{L} w_l a_l(x)$. In the experiments, we simply set all the weight coefficients $w_l$ to 1, i.e., the aggregated design criterion is the sum of individual property constraints.

### B.3 BASELINES

We briefly overview the existing methods and baselines that we have used for all experiments. Some baselines are applicable for more than one experiment while some specialize in a certain experiment.

**JT-VAE (Jin et al., 2018)**  The junction tree variational autoencoder (JT-VAE) reconstructs a molecule in its graph (2 dimensional) representation using its junction tree and molecular graph as inputs. JT-VAE learns a structured, fixed-dimensional latent space. During inference, controllable generation is achieved by first performing optimization on the latent space via property predictors trained on the latent space and then generating a molecule via the decoder in the VAE.

**MMPA (Dalke et al., 2018)**  The Matched Molecular Pair Analysis (MMPA) platform uses rules and heuristics, such as matched molecular pair, to perform various molecule operations such as search, transformations, and synthesizing.

**GCPN (You et al., 2018)**  Graph Convolutional Policy Network (GCPN) trains a graph convolutional neural network to generate molecules in their 2D graph representations with policy gradient reinforcement learning. The reward function consists of task-specific property predictors and adversarial loss.

**Vseq2seq (Bahdanau et al., 2015)**  This is a basic sequence-to-sequence model borrowed from the machine translation literature that translates the input molecule to an output molecule with more desired properties.

**VJTNN (Jin et al., 2019)**  VJTNN is a graph-based method for generating molecule graphs based on junction tree variational autoencoders (Jin et al., 2018). The method formalizes the controllable molecule generation problem as a graph translation problem and is trained using an adversarial loss to match the generation and data distribution.

**HierG2G (Jin et al., 2019)**  The Hierarchical Graph-to-Graph generation method takes into account the molecule's substructures, which are interleaved with the molecule's atoms for more structured generation.

**AtomG2G (Jin et al., 2019)**  The Atom Graph-to-Graph generation method is similar to HierG2G but only takes into account of the molecule's atoms information without the structure and substructure information.

**DESMILES (Maragakis et al., 2020)**  The DESMILES method aims to translate the molecule's fingerprint into its SMILES representation. Controllable generation is achieved by fine-tuning the model for task-specific properties and datasets.

**MolDQN (Zhou et al., 2019)**  The Molecule Deep Q-Learning Network formalizes molecule generation and optimization as a sequence of Markov decision processes, which the authors use deep Q-learning and randomized reward function to optimize.

**GA (Nigam et al., 2019)**  This method augments the classic genetic algorithm with a neural network which acts as a discriminator to improve the diversity of the generation during inference.

**QMO (Hoffman et al., 2021)**  The Query-based Molecule Optimization framework is a latent-optimization-based controllable generation method operated on the SMILES representation of molecules. Instead of finding a latent code in the latent space via property predictor trained on the latent space, QMO uses property predictor on the molecule space and performs latent optimization via zeroth order gradient estimation. Although this latent-optimization-based method removes the need to train latent-space property predictors, we find that it is sometimes challenging to tune the hyper-parameters to adapt QMO for different controllable generation tasks. For example, in the SARS-CoV-2 main protease inhibitor design task, we could not succeed to generate an optimized molecule using QMO even with extensive hyper-parameter tuning.

**GVAE-RL (Jin et al., 2020a)**  This method is the graph-based grammar VAE (Kusner et al., 2017) that learns to expand a molecule with a set of expansion rules, based on a variational autoencoder. GVAE-RL further fine-tunes GVAE with RL objective for controllable generation, using the property values as the reward.

**REINVENT (Olivecrona et al., 2017)**  REINVENT trains a recurrent neural network using RL techniques to generate new molecules.

**RationaleRL (Jin et al., 2020a)**  The RationalRL method first assembles a "rationale", a molecule fragment composed from different molecules with the desired properties. Then it trains a decoder using the assembled collection of rationales as input by first randomly sampling from the decoder, scoring each sample with the property predictors, and use the positive and negative samles as training data to fine-tune the decoder.

**MARS (Xie et al., 2021)**  The MARS method models the molecule generation process as a Markov chain, where the transitions are the "edits" on the current molecule graph parametrized by a graph neural network. The generation process proceeds by either accepting the newly edited molecule if it has more desired attributes than the previous one. Otherwise, the previous molecule is kept and the Markov chain continues.

**Graph MCTS and Graph GA (Jensen, 2019)**  Graph MCTS and GA methods traverse the molecule space in its graph representation using genetic algorithm and Monte Carlo Tree Search algorithms, respectively.

**SMILES LSTM (Segler et al., 2017)**  This method is a decoder (an LSTM) only method which generates molecules using a seed input symbol. Controllable generation is achieved by fine-tuning it on a task-specific dataset with RL using an property scores as the reward function.

**SMILES GA (Yoshikawa et al., 2018)**  This method, similar to GA, applies various rules to the SMILES representation of molecules from a starting population.

### B.4  QED AND PENALIZED LOGP EXPERIMENTS

To ensure a fair comparison with existing methods, we rely on using the same total number of calls to the property predictors. For example, an optimization process for an input molecule that runs for $T$ iterations with $M$ calls to the property predictors at each iteration will invoke a total of $T \times M$ property predictor calls. We use the same or less number of property predictor calls for each molecule's optimization.

**QED experiment setup.**  When running RetMol, for each input molecule, we set the maximum number of iterations to 1000 and sample 50 molecules at each iteration, resulting in a total of $1000 \times 50 = 50,000$ property predictor calls. This number matches that in QMO (Hoffman et al., 2021), where the maximum number of optimization iterations is 20 with 50 "restarts" and 50 samples at each iteration for gradient estimation. Effectively, this results in a total of $20 \times 50 \times 50 = 50,000$

Table A1: Number of generated molecules (or number of calls to the property prediction) for the competitive methods in the task of optimizing four properties: QED, SA, and two binding affinities to GSK3$\beta$ and JNK3 estimated by pre-trained models from (Jin et al., 2020a).

| Method | Number of generated samples |
|---|---|
| RationaleRL (Jin et al., 2020a) | 1,086,000 |
| MARS (Xie et al., 2021) | 2,750,000 |
| MolEvol (Chen* et al., 2021) | 200,000 |
| **RetMol** | 296,000 |

calls to the scoring function, which is the same as in our setting. We evaluate performance by success rate. For each input molecule, if a molecule that satisfy the design criteria (QED is above 0.9 and similarity with the input molecule is above 0.4), then we count it as a success. Success rate for all 800 input molecules is thus

$$\text{success rate} = \frac{\#\text{successful input molecules}}{800}.$$

**Penalized logP experiment setup.** For each input molecule, we run the optimization for 80 iterations and sample 100 molecules at each iteration, which is exactly the same setting as in QMO (Hoffman et al., 2021). If optimization fails, i.e., no new molecules are generated that have higher penalized logP value than the input, then we set the relative improvement to 0. We evaluate performance by difference in penalized logP between the optimized and the input molecules, averaged over all 800 input molecules:

$$\text{avg. improvement} = \frac{1}{800} \sum_{i=1}^{800} \left( a_{\text{plogP}}(x'^{(i)}) - a_{\text{plogP}}(x_{\text{in}}^{(i)}) \right),$$

where $a_{\text{plogP}}$ is the penalized logP predictor, $x'$ is the generated molecule and $x_{\text{in}}$ is the input molecule.

### B.5  GSK3$\beta$ + JNK3 + QED + SA EXPERIMENT SETUP

For a fair comparison, we follow the same setup as in (Jin et al., 2020a; 2018) and make two major changes to RetMol's generative process. First, rather than starting with a known molecule, we first draw a random latent embedding as the starting point of the molecule generation process. 2) In the iterative refinement process, we generate only one molecule and directly use it as the input molecule in the next iteration; that is, in this experiment, at each iteration, we do not use the property predictors to select the best generated molecules because there is only one generated per iteration. For each input molecule, we run the optimization for 80 steps and generate one molecule at each step following (Jin et al., 2018). Doing so results in a total of $80 \times 1 \times 3,700 = 296,000$ generated molecules, which is an order of magnitude less than the $550 \times 5,000 = 2,750,000$ number of generated molecules required for MARS (Xie et al., 2021). This number in MARS remains high even if we change the number of samples at each iteration from $5,000$ to $3,700$ to align with the number of molecules evaluated in RetMol and the other baselines. In Table A1, we provide the complete comparison of the competitive methods regarding the number of generated samples, where we can see that our method is sample efficient (with the best performance shown in Table 1b).

Unlike the previous tasks, there is no specification as to which molecule to use as the input molecule to be optimized. To obtain the input molecules, we first randomly select 3700 molecules from the CheMBL dataset (Gaulton et al., 2016), retrieve exemplar molecule randomly from the entire CheMBL dataset (Gaulton et al., 2016), and greedily generate one molecule using the random input and random retrieved exemplar molecules. This one generated molecule is the input to RetMol. We choose 3700 molecules to be optimized because 3700 is the number of molecules evaluated in existing methods reported in (Jin et al., 2020a). Note that the inputs to RetMol differ from those in existing methods. However, we believe our choice of input does not put our method in an advantage over existing methods and may even be at an disadvantage compared to existing methods. For example, the inputs to RationaleRL (Jin et al., 2020a) are "rationales", i.e., molecule fragments that already satisfy either the GSK3$\beta$ or the JNK3 property and that are pieced together. These rationales are usually very close to the desired molecules. In contrast, the inputs to RetMol are mostly molecules with very low (close to 0) GSK3$\beta$ and JNK3 values, making the optimization challenging.

Below, we show how to compute the metrics when using RetMol, which largely follows the computation in (Jin et al., 2018; 2020a). For success rate, we count number of input molecules that result in at least one successful generated molecule (satisfy the four property constraints simultaneously) from all 80 molecules generated for that input molecule. If there exist more than one successful molecule for a given input, we choose the one with the least Tanimoto distance (Bajusz et al., 2015) with the input for the evaluation in the remaining metrics. For Novelty, we compute the percentage of the selected successful molecules that have less than 0.4 Tanimoto distance with any molecules in the retrieval database, i.e., those molecules that already satisfy the property constraints. For diversity, we compute the pairwise Tanimoto distance between each pair of the selected successful molecules.

## B.6 GUACAMOL BENCHMARK EXPERIMENT SETUP

There are no known input molecules to any tasks in the Guacamol benchmark. Therefore, we first randomly select a population of molecules from the CheMBL dataset, which is the standard dataset in the benchmark. In this work, we choose five molecules as the starting population, although a larger population size is likely to yield better results at the higher computational cost. We perform iterative refinement for each molecule in the input population for 1000 iterations. After the optimization ends, we collect all the generated molecules from all molecules in the population and select the best $N$ molecules, based on which the benchmark score is computed for each benchmark task. In all the MPO tasks in the benchmark, $N = 100$. The properties to be optimized and the scoring function details for each benchmark task are available in (Brown et al., 2019) and at `https://github.com/BenevolentAI/guacamol`. The scoring functions and evaluation protocol are provided in the benchmark. The benchmark provide an API which takes a population of input molecules and scoring functions to generated the optimized molecules. We implement RetMol using their API, which ensures a fair comparison and evaluation using the same evaluation protocol defined in the Guacamol benchmark.

## B.7 SARS-CoV-2 MAIN PROTEASE INHIBITOR DESIGN EXPERIMENT SETUP

For RetMol, for each of the eight input inhibitors to be optimized, we run the iterative refinement for 10 iterations with 100 samples per iteration. For the graph GA baseline, for each of the eight inhibitors to be optimized, we use the same inhibitor as the initial population for crossover and mutation. We set the population and offspring size to 100 and run it for 10 iterations.

We use binding affinity between the generated inhibitor and the protein target. We computationally approximate binding affinity using Autodock-GPU, [4] which significantly speeds up docking compared to its CPU variants. The input files contains the receptor (target protein) of interest, the generated or identified ligands (inhibitor molecules), and a bounding box that indicates the docking site. For different docking experiments, the receptor and bounding box need to be prepared differently. Once the receptor and bounding box are prepared for a particular docking experiment, they are kept fixed and used for docking all ligands throughout the experiment. For the SARS-CoV-2 main protease docking experiments, we use the protein with PDB ID 7L11. We choose the bounding box by first docking a the protein with a known inhibitor ligand, Compound 5 (Zhang et al., 2021), and then extract the bounding box configurations from its best docking pose (Gao et al., 2022). The receptor, ligand, and bounding box preprocessing steps use the Autodock software suite.[5]

In addition, as we mentioned in the main paper, we also tested on QMO (Hoffman et al., 2021) with the following configurations: we set optimization iterations to 100, number of restarts to 1, weights for the property (binding affinity) to be optimized in $\{0.005, 0.01, 0.1, 0.25\}$, base learning rate in $\{0.05, 0.1\}$, and the property constraints the same as in those in RetMol. In our experiments, these configurations did not succeed in generating an optimized molecule given the constraints. We follow the official QMO codebase in these experiments. [6]

---

[4] `https://github.com/ccsb-scripps/AutoDock-GPU`
[5] `https://autodock.scripps.edu/`
[6] `https://github.com/IBM/QMO`

### B.8   ANALYSES EXPERIMENTS: TRAINING OBJECTIVES

The purpose of this experiment is two-fold: 1) We want to show if only updating the fusion module while fixing the weights of the base generative model during training will cause a degradation in the quality of generated molecules; and 2) we want to check if our proposed nearest neighbor objective can achieve better generation quality than the conventional language modeling objective (i.e., predicting the next token of input).

To evaluate RetMol, the unconditional generation procedure is the same as that in training the information fusion module, i.e., the retrieval database is the training split of the ZINC250 dataset, and the retrieval criterion is molecule similarity. Besides, since we mainly focus on evaluating the quality of generated molecules, we thus use validity, novelty and uniqueness as our metrics, which are standard in literature for evaluating molecule generation models' performance. The results suggest that our training objective that only updates the fusion module does not sacrifice the unconditioned molecule generation performance and even results in slight improvement on the novelty metric.

We perform the evaluation on the test split of the ZINC250k dataset. For each molecule in the test set, we generate 10 molecules by first randomly perturb the input to the decoder, i.e., an encoded (or fused) embedding matrix, 10 independent times using a isotropic random Gaussian with standard deviation of 1. Then, for validity, we compute the percentage of the 10 generated molecules that are valid according to RDKit, [7] averaged over all test molecules. For novelty, we compute the percentage of the 10 generated molecules that are not in the training split of the ZINC250k dataset, averaged over all molecules in the test set. For uniqueness, we compute the percentage of the 10 generated molecules that are unique, averaged over all molecules in the test set.

### B.9   REMARKS ON NUMBER OF ITERATIONS

In most experiments, we keep the number of iterative refinements in RetMol the same as the baseline iterative methods. For example, in Section 3.1, we use 80 optimization iteration steps for RetMol, which is the same as QMO. In Section 3.2, we use 80 optimization iteration steps, which is the same as JT-VAE. Note that in this experiment, RationaleRL does not require such iterative refinement process but does require extensive task-specific training and fine-tuning. In Section 3.3, the benchmark results do not mention the number of iterations and therefore we simply set the max number of iterations to 1k for RetMol. In Section 3.4, we use 20 iterations for both RetMol and Graph GA.

Our results also suggest that one does not need as many iterations as the baselines to achieve strong results. For example, we showed in Figure 3 (middle) that for experiments in Section 3.2, RetMol achieves better results than baselines with only 30 iterations.

## C   ADDITIONAL EXPERIMENT RESULTS AND ANALYSES

### C.1   QED AND LOGP EXPERIMENT

To visualize the property value improvements, we plot in Figure A1 (i) the QED of the optimized (generated) molecules and of the input molecules under similarity constraint $\delta = 0.4$ and (ii) the penalized logP improvement between the generated and the input molecules under similarity constraints $\delta = \{0.4, 0.6\}$.

We can see that, for the penalized logP experiment, for similarity constraint $\delta = 0.4$, there are quite a few molecules with a large penalized logP value improvements. These molecules are the reason that the variance for RetMol in Table 1b is large. But even if we remove those molecules with extreme values, i.e., penalized logP bigger than 20, we still obtain an average improvement of $8.17 \pm 4.12$, which is still better than the best existing method. Furthermore, we also compute the mode of the penalized logP values for our method (i.e., the $x$-axis value of the largest peak in Figure A1, Right), and we get 3.804 for $\sigma = 0.6$ and 6.389 for $\sigma = 0.4$. We can see our mode of the penalized logP values is still better than the mean in most of the baselines in Table 1b. For QMO, their mode looks very similar to ours from Figure 8 in their paper (Hoffman et al., 2021).

---

[7]https://www.rdkit.org/

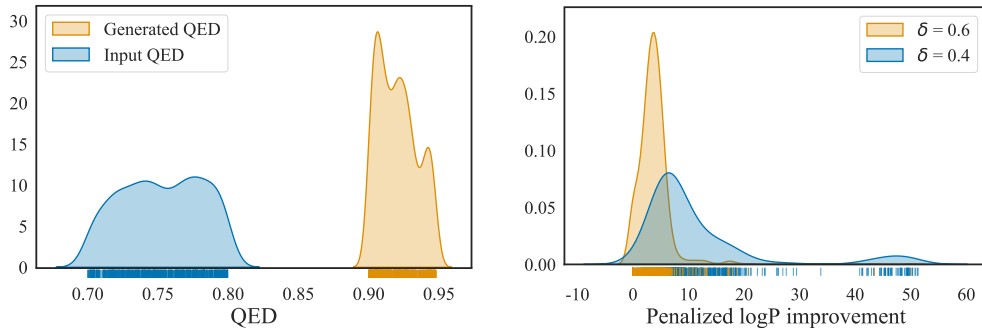

Figure A1: **Left**: distribution of QED values of the original and the optimized (generated) molecules under similarity constraint $\delta = 0.4$. **Right**: distribution of penalized logP improvement comparing similarity constraints $\delta = \{0.4, 0.6\}$.

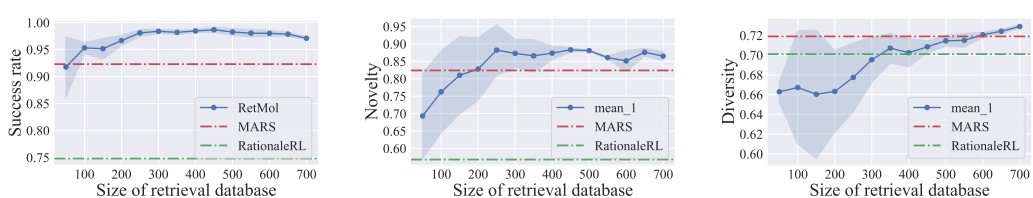

Figure A2: Generation performance with varying retrieval database size on the experiment in Section 3.2. Our framework achieves strong performance with as few as 100 molecules in the retrieval database and performance generally improves with increasing retrieval database size on all metrics.

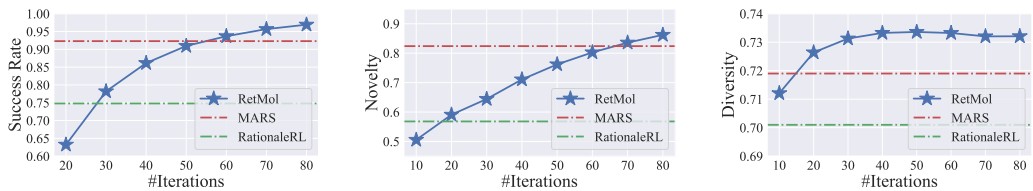

Figure A3: Generation performance with varying number of optimization iterations on the experiments in Section 3.2. Dashed lines are the success rates and novelty scores of the two best baselines. We observe that all the metrics improve as we increase the number of iterations.

## C.2   GSK3$\beta$ AND JNK3 EXPERIMENT

In Figure A4, we show 54 randomly chosen molecules generated by RetMol along with their highest similarity with the molecules in the retrieval database. These molecules demonstrate the diversity and novelty of the molecules that RetMol is capable of generating.

## C.3   GUACAMOL EXPERIMENT

We show in Table A2 the detailed benchmark, SA, and QED results for all the MPO tasks in the Guacamol benchmark.

We additionally apply the functional group filters [8] to the optimized molecules of each method and compare the average number of molecules that pass the filters. Figure A5 visualizes the results, which align with those in Sec. 3.3: RetMol strikes the best balance between optimizing benchmark score and maintaining a good functional group relevance. For example, Graph GA achieves the best benchmark performance but fails the filters more often than some of the methods under comparison. In contrast, RetMol achieves the second best benchmark performance and the highest number of

---

[8] https://github.com/PatWalters/rd_filters

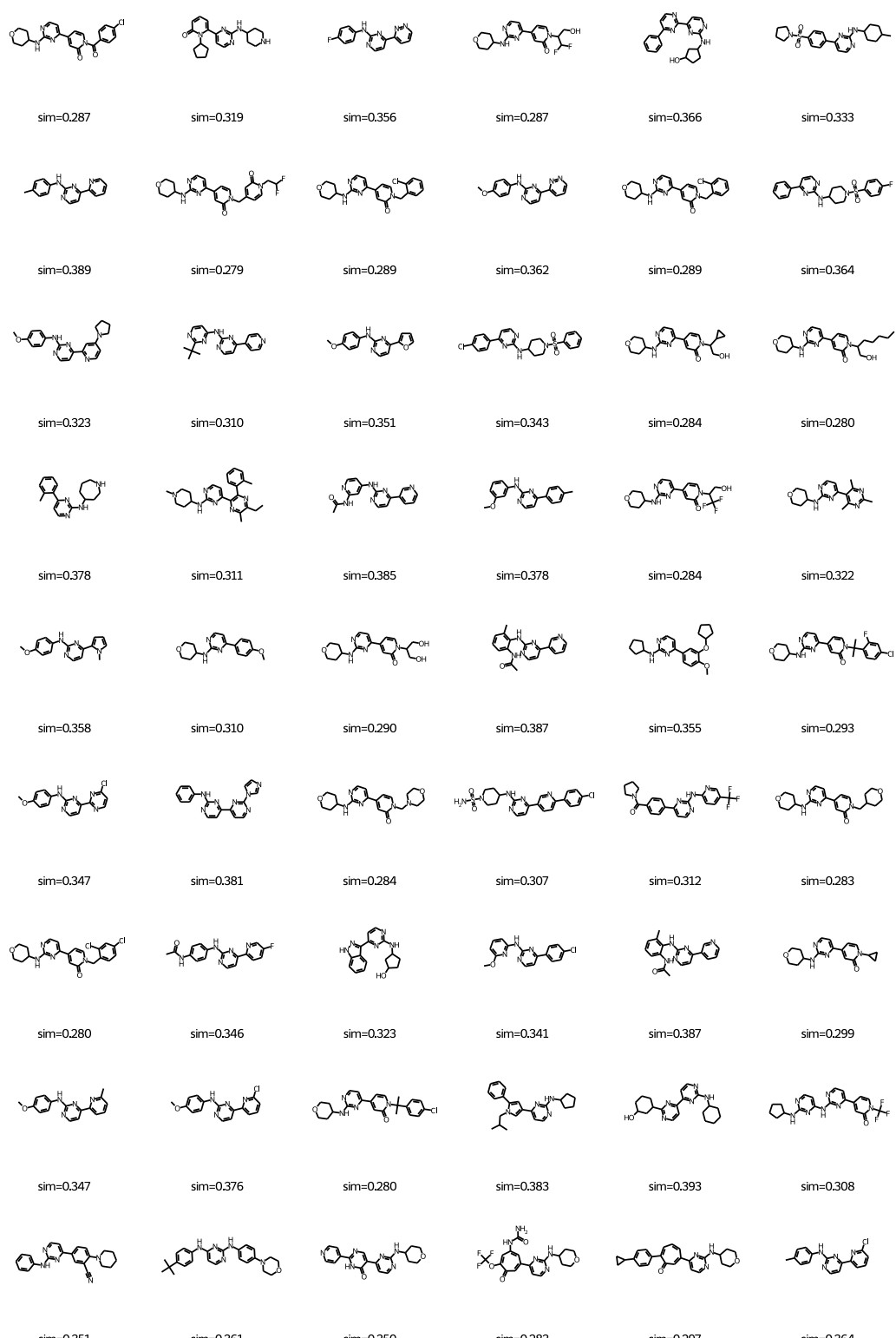

Figure A4: Visualizations of randomly chosen molecules generated by RetMol for the GSK3$\beta$ + JNK3 + QED + SA experiment. Below each generated molecule, we show its highest similarity between each molecule in the retrieval database.

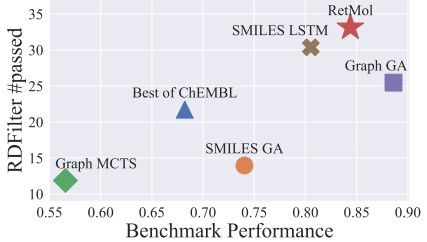

Figure A5: Average number of optimized molecules that pass the functional group filters against Guacamol benchmark scores. The results align with those in Sec. 3.3: RetMol strikes the best balance between optimizing benchmark score and maintaining a good functional group relevance.

Table A2: Detailed results from the Guacamol MPO results. The tables from the top to the bottom are the benchmark results, averaged SA values, averaged QED values, and averaged numbers of generated molecules that pass the functional group filters, respectively. SA and QED values and the number of filter-passing molecules are averaged over all the 100 molecules evaluated in each MPO task. Bold and underline represent the best and the second best in each metric in each benchmark task, respectively.

| Benchmarks | SMILES GA | Graph MCTS | Graph GA | SMILES LSTM | Best of Dataset | Ours |
|---|---|---|---|---|---|---|
| Osimertinib MPO | 0.886 | 0.784 | **0.953** | 0.907 | 0.839 | 0.915 |
| Fexofenadine MPO | 0.931 | 0.695 | **0.998** | 0.959 | 0.817 | 0.969 |
| Ranolazine MPO | 0.881 | 0.616 | 0.920 | 0.855 | 0.792 | **0.931** |
| Perindopril MPO | 0.661 | 0.385 | 0.792 | **0.808** | 0.575 | 0.765 |
| Amlodipine MPO | 0.722 | 0.533 | **0.894** | **0.894** | 0.696 | 0.879 |
| Sitagliptin MPO | 0.689 | 0.458 | **0.891** | 0.545 | 0.509 | 0.735 |
| Zaleplon MPO | 0.413 | 0.488 | **0.754** | 0.669 | 0.547 | 0.713 |

| Benchmarks - SA | SMILES GA | Graph MCTS | Graph GA | SMILES LSTM | Best of Dataset | Ours |
|---|---|---|---|---|---|---|
| Osimertinib MPO | 6.386 | 3.901 | 3.357 | 2.923 | **2.705** | 3.061 |
| Fexofenadine MPO | 3.590 | 4.671 | 3.897 | 3.171 | **3.097** | 3.546 |
| Ranolazine MPO | 6.071 | 4.110 | 4.111 | **2.900** | 3.226 | 3.456 |
| Perindopril MPO | 5.343 | **3.365** | 4.286 | 4.017 | 3.645 | 4.276 |
| Amlodipine MPO | 4.717 | 3.529 | 3.575 | 3.329 | **3.163** | 3.431 |
| Sitagliptin MPO | 6.743 | 5.183 | 6.804 | **2.794** | 2.886 | 3.715 |
| Zaleplon MPO | 3.244 | 3.216 | 2.899 | 2.387 | **2.294** | 2.622 |

| Benchmarks - QED | SMILES GA | Graph MCTS | Graph GA | SMILES LSTM | Best of Dataset | Ours |
|---|---|---|---|---|---|---|
| Osimertinib MPO | 0.256 | 0.443 | 0.197 | 0.240 | **0.478** | 0.264 |
| Fexofenadine MPO | 0.207 | 0.495 | 0.309 | 0.335 | **0.382** | 0.301 |
| Ranolazine MPO | 0.096 | **0.305** | 0.095 | 0.113 | 0.129 | 0.112 |
| Perindopril MPO | 0.481 | 0.477 | 0.365 | 0.465 | 0.421 | **0.546** |
| Amlodipine MPO | 0.146 | **0.582** | 0.351 | 0.386 | 0.472 | 0.365 |
| Sitagliptin MPO | 0.254 | 0.453 | 0.086 | 0.700 | **0.735** | 0.701 |
| Zaleplon MPO | 0.206 | 0.679 | 0.562 | **0.730** | 0.712 | 0.753 |

| Benchmarks - filters | SMILES GA | Graph MCTS | Graph GA | SMILES LSTM | Best of Dataset | Ours |
|---|---|---|---|---|---|---|
| Osimertinib MPO | 0 | **23** | 0 | 0 | 19 | 0 |
| Fexofenadine MPO | 58 | 22 | **73** | 68 | 47 | 43 |
| Ranolazine MPO | 0 | 0 | 0 | 0 | 0 | 0 |
| Perindopril MPO | 12 | 29 | 42 | 60 | 40 | **90** |
| Amlodipine MPO | 4 | 31 | 56 | **78** | 49 | 58 |
| Sitagliptin MPO | 0 | 6 | 0 | 69 | 76 | **82** |
| Zaleplon MPO | 0 | 51 | 37 | 97 | 84 | **98** |

passed molecules. Please see Table A2 for the number of optimized molecules that pass the filters for each MPO task and for each method.

Table A3: Visualizations of the original and optimized inhibitors from RetMol for the SARS-CoV-2 main protease. Similarity constraint here is $\delta = 0.6$.

| Inhibitor name | Original | Properties (original) | | Optimized | Properties (optimized) | | Similarity map | Similarity |
|---|---|---|---|---|---|---|---|---|
| Favipiravir | | Docking | -4.93 | | Docking | -6.78 | | 0.60 |
| | | QED | 0.55 | | QED | 0.77 | | |
| | | SA | 2.90 | | SA | 3.50 | | |
| Bromhexine | | Docking | -9.64 | | Docking | -11.48 | | 0.60 |
| | | QED | 0.78 | | QED | 0.64 | | |
| | | SA | 2.94 | | SA | 2.57 | | |
| PX-12 | | Docking | -6.13 | | Docking | -8.45 | | 0.65 |
| | | QED | 0.74 | | QED | 0.64 | | |
| | | SA | 3.98 | | SA | 3.80 | | |
| Disulfiram | | Docking | -8.58 | | Docking | -9.09 | | 0.64 |
| | | QED | 0.57 | | QED | 0.60 | | |
| | | SA | 3.12 | | SA | 3.39 | | |
| Kaempferol | | Docking | -8.45 | | Docking | -8.54 | | 0.62 |
| | | QED | 0.55 | | QED | 0.63 | | |
| | | SA | 2.37 | | SA | 2.47 | | |

## C.4 SARS-COV-2 MAIN PROTEASE INHIBITOR DESIGN EXPERIMENT

Tables A3 and A4 visualizes the original and the RetMol optimized inhibitors along with the similarity map, Tanimoto distance, QED, SA, and docking (unit in `kcal/mol`) scores for both similarity constraint $\delta = \{0.6, 0.4\}$. We highlight the properties that the initial inhibitor do not satisfy in red and the same properties in the optimized inhibitor in green. We can see that RetMol not only optimizes the docking score but also successfully improves the QED and SA scores for some inhibitors.

## C.5 ANTIBACTERIAL DRUG DESIGN FOR THE MURD PROTEIN

In addition to the experiments above and in the main paper, here we demonstrate another real-world use case of RetMol for Antibacterial drug design. We choose the MurD protein (PDB ID: 3UAG) as the target. This is a promising target for antibiotic design because it is necessary for the development of the cell wall, which is essential to the bacterial survival. Inhibiting this target thus has the potential to destroy the bacterial without harming humans because the cell wall and thus the target protein is absent in animals (Sangshetti et al., 2017). [9]

The design criteria in this controllable generation experiment is similar to those in the SARS-CoV-2 main protease inhibitor design experiment. In addition to improving the binding affinity of selected weakly-binding molecules, we have several desired properties [10] including a small logP value (below 3), a large QED value (above 0.6), a small SA score (below 4), and a molecule weight between 250-350 Da. Since the task encourages diverse generations, we set a small similarity threshold to 0.2. We select 100 molecules from bindingDB (Liu et al., 2007) that has experimental binding values to MurD, and choose eight molecules with the lowest binding affinity as input to be optimized. For

---

[9] Also see here: `https://github.com/opensourceantibiotics/murligase/wiki/Overview`

[10] `https://github.com/opensourceantibiotics/murligase/issues/69`

Table A4: Visualizations of the original and optimized inhibitors from RetMol for the SARS-CoV-2 main protease. Similarity constraint here is $\delta = 0.4$.

| Inhibitor name | Original | Properties (original) | | Optimized | Properties (optimized) | | Similarity map | Similarity |
|---|---|---|---|---|---|---|---|---|
| Favipiravir | | Docking | -4.93 | | Docking | -8.7 | | 0.41 |
| | | QED | 0.55 | | QED | 0.62 | | |
| | | SA | 2.90 | | SA | 3.25 | | |
| Bromhexine | | Docking | -9.64 | | Docking | -12.65 | | 0.40 |
| | | QED | 0.78 | | QED | 0.63 | | |
| | | SA | 2.38 | | SA | 2.48 | | |
| PX-12 | | Docking | -6.13 | | Docking | -10.90 | | 0.50 |
| | | QED | 0.74 | | QED | 0.62 | | |
| | | SA | 3.98 | | SA | 3.72 | | |
| Ebselen | | Docking | -7.31 | | Docking | -10.82 | | 0.40 |
| | | QED | 0.63 | | QED | 0.61 | | |
| | | SA | 2.05 | | SA | 2.27 | | |
| Disulfiram | | Docking | -8.58 | | Docking | -10.44 | | 0.45 |
| | | QED | 0.57 | | QED | 0.61 | | |
| | | SA | 3.12 | | SA | 3.63 | | |
| Entecavir | | Docking | -9.00 | | Docking | -12.34 | | 0.41 |
| | | QED | 0.53 | | QED | 0.64 | | |
| | | SA | 4.09 | | SA | 3.76 | | |
| Quercetin | | Docking | -9.25 | | Docking | -9.84 | | 0.41 |
| | | QED | 0.43 | | QED | 0.62 | | |
| | | SA | 2.54 | | SA | 2.62 | | |
| Kaempferol | | Docking | -8.45 | | Docking | -10.35 | | 0.41 |
| | | QED | 0.55 | | QED | 0.67 | | |
| | | SA | 2.37 | | SA | 2.51 | | |

Table A5: Antibacterial drug design with the MurD target comparing RetMol with Graph GA. RetMol optimizes input molecules better (in terms of binding affinity, unit in `kcal/mol`) than Graph GA under various property constraints.

| Input | Input score | RetMol optimized | Graph GA (Jensen, 2019) optimized |
|---|---|---|---|
| Oc1c2SCCc2nn1-c1cccc(Cl)c1 | -7.73 | -11.78 | -9.76 |
| Oc1c2SCCc2nn1-c1ccc(cc1)C(F)(F)F | -7.31 | -12.82 | -11.09 |
| Oc1c2SCCc2nn1-c1ccc(Cl)cc1 | -7.53 | -12.46 | -9.31 |
| CC1Cc2nn(c(O)c2S1)-c1ccc(Cl)cc1 | -7.86 | -13.50 | -9.39 |
| Oc1c2SCCc2nn1-c1cccc(c1)C(F)(F)F | -7.74 | -14.72 | -8.98 |
| Oc1c2SCCc2nn1-c1ccccc1C(F)(F)F | -7.63 | -13.62 | -8.81 |
| Oc1c2SCCc2nn1-c1ccccc1 | -7.70 | -14.00 | -10.68 |
| Oc1c2SCCc2nn1-c1ccc(F)cc1 | -7.19 | -13.05 | -11.67 |
| **Average improvement** | - | **5.66** | 2.38 |

RetMol, we use all the molecules resulted from bindingDB as the retrieval database. The remaining experiment procedures, including docking, follows from those in the SARS-CoV-2 main protease inhibitor design experiment. Table A5 compares the generation performance of RetMol with Graph GA. We can see that RetMol optimizes the input molecules better than Graph GA.

## C.6 ANALYSES

**With and without retrieval module.** We test the base generative model in our framework on the experiment in Section 3.2 with the same setup as our full framework. The base generative model without the retrieval module is unable to generate any molecules that satisfy the given constraints demonstrating the importance of the retrieval module in achieving controllable generation.

**Varying retrieval database size.** Figure A2 shows how RetMol performs with varying retrieval database size with the novelty (middle) and diversity (right) metrics in addition to the success rate metric (left). We can observe that, similar to success rate, RetMol can achieve reasonable novelty and diversity with a small retrieval database, albeit with larger variance. The performance continues to improve and the variance continues to decrease with a larger retrieval database. This experiment corroborate the analyses in the main paper that RetMol is efficient in the sense that a small retrieval database can already provide a strong performance.

**Varying refinement iterations.** Figure A3 shows how RetMol performs with respect to the novelty (middle) and diversity (right) metrics in addition to success rate (left). These results are similar to and corroborate those presented in the main paper: RetMol achieves strong results and beats the best existing method with as little as 10 iterations on the diversity metric and with as little as 20 iterations on the novelty metric. Performance also continues to improve with more iterations.

**Comparing to parameter-free information fusion** To show the effectiveness of our proposed information fusion module that involves additional trainable parameters from the cross entropy function, we introduce two parameter-free information fusion modules detailed below for comparison.

Both two methods need to calculate a property score for each of the retrieved molecules (and we take an average of the property scores as the final score in the case of multi-property optimization), and then apply the softmax function on these scores to obtain a vector of normalized scores. The first method, termed **softmax-aggregation**, applies a weighted averaging of the retrieved embeddings as the fused embedding, with the weights being set to the normalized scores. Note that since all retrieved embeddings have different lengths, we simply concatenate zero vectors to those embeddings with a smaller length to maintain the same dimensionality before the weighted averaging. The second method, termed **softmax-sampling**, samples one retrieved molecule embedding as the fused embedding, by treating the normalized scores from softmax as probabilities of a multinomial distribution. The rest procedures are kept the same with RetMol.

Table A6: We compare our proposed information fusion module with two parameter-free fusion methods in the QED experiments in Section 3.1, where the results demonstrate the importance of our proposed information fusion module.

| Method | Success (%) |
|---|---|
| Softmax-aggregate | 1.5 |
| Softmax-sampling | 53.6 |
| **RetMol** | **94.5** |

We summarize the results in Table A6, where we conduct the QED experiments in Section 3.1. Compared with RetMol (> 90% success rate), the softmax-aggregate method has nearly zero success rate. It implies that training the information fusion component is necessary and that simply aggregating the embeddings from the retrieved exemplar molecules results in undesirable results. The softmax-sampling method that does not directly aggregate the information also achieves subpar performance compared to RetMol, which also demonstrates the importance of our proposed information fusion module. These two comparisons together show that our proposed information fusion module with

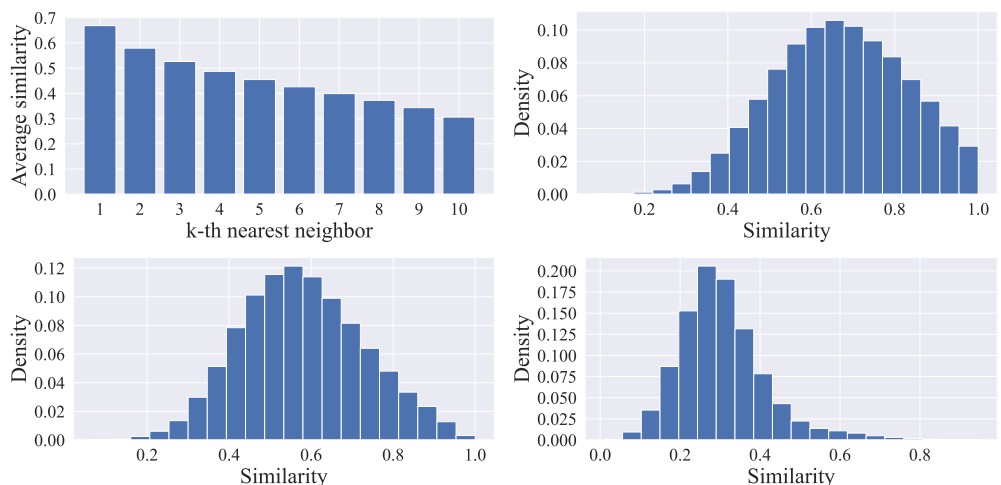

Figure A6: Analyses of the similarities between the retrieved molecules and the input molecules. **Top left**: the average similarity between the $k$-th most similar molecules to their corresponding input molecules; **Top right, bottom left, and bottom right**: the distribution of the similarities of the 1st, 2nd, and 10th most similar molecules to their corresponding input molecules, respectively.

trainable parameters that learns to dynamically aggregate retrieved information is critical for achieving good generation performance.

**Computational cost**   For the memory cost, 1) the model size (in #parameters) does not increase much: The information fusion module contains 460,544 parameters. The base generative model in RetMol contains 10,010,635 parameters. These numbers indicate that the information module adds only less than 5% of the total parameters. 2) The memory cost in the fusion module scales linearly with the number of retrieved molecules $K$ in order to store the $K$ extra embeddings of the retrieved molecules, i.e., $O(K\bar{L}D)$, where $\bar{L}$ is the average molecule length, and $D$ is the embedding size. Since $K$ is small (i.e., $K = 10$ with $\bar{L} = 55$, $D = 256$), in experiments we find this additional memory cost is small.

For the time cost, we also observed infinitesimal difference in the runtime between the base generative model (i.e., without the fusion module) and RetMol (i.e., with the fusion module). To verify this, we ran a small experiment to compare their time cost in the encoding step, as RetMol uses the same decoder as the base generative model. Specifically, we gave 100 input molecules to both the base generative model and RetMol with a batch size of 1 and compute their average runtime (in seconds) on NNVIDIA Quadro RTX 8000. The average encoding time for the base generative model and for RetMol is **0.00402** seconds and **0.00343** seconds, respectively. It shows that the additional time cost of the fusion module in RetMol is indeed small.

**Similarities of retrieved molecules**   We first plot in Figure A6 the similarities between the input molecule and each of its $K$ nearest neighbors (measured by cosine similarity) where $K = 10$, respectively, and take an average across all input molecules in the training set. We see that on average, the most similar molecule has a similarity of 0.66 to its corresponding input, and the second most similar molecule has a similarity of 0.57. The average similarities of the remaining 3rd to the 10th most similar molecules do not drop dramatically, implying the retrieved molecules are indeed similar ones to the input in the case of $K = 10$. Second, we also plot in Figure A6 the distribution of similarities for the 1st, 2nd, and 10th retrieved molecules to their input molecule, respectively, across the training set. We see that the majority of the top-$k$ similar molecules have a similarity greater than 0.2 with their input, even for the distribution of the 10th retrieved molecules. It implies the possibility of retrieving a largely dissimilar molecule is small. These results both imply that the noise present in the retrieved molecules is modest with a small fixed $K$.

**Attribute relaxation ordering in molecule retriever**   In molecule retriever, we may need to gradually relax attribute constraints to construct a relaxed feasible set. Our strategy is to relax the

Table A7: Penalized logP experiment with HierVAE as the base molecule generative model (encoder and decoder) in the RetMol framework. This result demonstrates that RetMol is flexible and is compatible with models other than transformer-based ones and that the RetMol framework improves controllable molecule generation performance compared to the base model alone.

| Method | $\delta$=0.6 | $\delta$=0.4 |
|---|---|---|
| HierG2G | 2.49±1.09 | 3.98±1.46 |
| **RetMol (w/ HierG2G)** | **3.09±3.29** | **6.25±5.76** |

harder-to-satisfy constraints first and then the easier-to-satisfy ones. In other words, we first focus more on the simple attributes and then on the hard attributes (e.g., similarity $\rightarrow$ QED $\rightarrow$ docking), which gradually increases the level of inference difficulty over iterations. The order of the constraints being removed is an important consideration during inference, especially when there are many of them. To show this, we perform a sanity check on the impact of attribute relaxation ordering in the QED task in Sec 3.1, where, for a quick experiment, we use a subset of 200 molecules and 20 iterations. We see that if we first relax QED and then similarity, we get 89.5% success rate, and if we first relax similarity and then QED, we get 78.5% success rate. It confirms our intuition that it is preferable to relax the harder-to-satisfy constraints first and then the easier-to-satisfy ones.

## C.7 RETMOL WITH OTHER BASE GENERATIVE MODELS

We conduct all experiments in this paper until this point with a pre-trained molecule generative model based on the transformer (i.e., BART) architecture as the encoder and decoder model in the RetMol framework. As a further demonstration that RetMol is flexible and is compatible with other types of molecule generative models, and thanks to the reviewers' suggestions, we conduct the penalized logP experiment with HierG2G Jin et al. (2019) as the RetMol encoder and decoder models. The results in Table A7 suggest that the RetMol framework can also elevate the performance of graph-based generative models, i.e., HierG2G, on controllable molecule generation.

