# OpenReview forum: "Retrieval-based Controllable Molecule Generation"
_ICLR.cc/2023/Conference — ICLR 2023 notable top 25%_

### Official Review · Reviewer_3hQB · 2022-10-23

**Confidence:** 4
**Correctness:** 3
**Technical Novelty And Significance:** 3
**Empirical Novelty And Significance:** Not applicable
**Recommendation:** 8

**Clarity, Quality, Novelty And Reproducibility:**

**Clarity**

This work is extremely well written. It was a pleasure to read from top to bottom.

**Quality**

The quality of this work is high. The choice of experiments is sensible, several ablations and analyses are presented to understand the impact of the modeling choices, all the claims made are substantiated.

**Novelty**

The paper applies well-known ideas from retrieval-augmented language models to the space of molecules, which does qualify as novel to me. Plus, the nearest-neighbor training objective is a novel idea (to my knowledge).

**Reproducibility**

Not reproducible. The authors state that the code will be released in the future (upon approval, I suspect).

**Strength And Weaknesses:**

**Strenghts**

- Strong performances in what appears to be a well-designed set of experiments;
- Training is very efficient (just the retrieval module); inference is not affected too much since we are adding a small number of parameters with respect to the large size of the encoder-decoder backbone;
- Works with a small set of retrieved molecules (up to 23 in the experiments), which is ideal when one wants to generate alternative molecules from limited data;
- The nearest neighbor training objective is a simple but very clever idea that works experimentally.

**Weaknesses**

- Not sure it would be applicable in cases where no exemplar molecules to be retrieved are unavailable (as stated in Conclusions); this might limit its widespread applicability;
- It poorly exploits the chemical space, i.e. it will most likely generate molecules that are very similar to the input molecule.
- Even though it is claimed to work with any encoder-decoder architecture, performances are shown considering only a single backbone model (ChemFormer). I would have expected that its cross-applicability would be one of the main focuses of this paper, but unfortunately it is not.

**Summary Of The Paper:**

The work presents a retrieval-based framework for controllable molecule generation. The framework consists of:

- A mechanism to retrieve a set of molecules that (partially) satisfy a set of desired properties the generated molecule should have (e.g. similarity to the input molecule, enhanced drug-like properties);
- A fusion module that combines the retrieved molecules with the input molecule through cross-attention;
- A novel self-supervised objective to train the fusion module, consisting in reconstructing the nearest neighbor molecule of the input molecule (as opposed to the classical self-reconstruction objective);
- An iterative refinement process to update the retrieval database with the generated molecules.

The fusion module can be plugged-in into any pre-trained encoder-decoder architecture (in this work, they use the SMILES representation of molecules and the ChemFormer model), which is frozen (i.e. only the fusion model is trained for efficiency).

The proposed approach is evaluated in three controllable generation scenarios, one real-world drug design scenario (find novel inhibitors for SARS-COVID protease), and the GuacaMol benchmark. In every case, it obtains an improvement over a range of different metrics such as success rate or docking score.

**Summary Of The Review:**

This appears to me as a strong submission. I am giving a score of 6 for the time being. Will be happy to raise my score when the authors answer the following set of questions/comments:

- In section 2, you acknowledge the existence of weighting coefficients $w_{\ell}$. However, they don't seem to be present in the rest of the work. What are they (or, are they just construction to explain the problem)?
- What is the justification for having used ChemFormer instead of any other encoder-decoder architecture as the backbone model? It would be great to see RetMol applied to another backbone, to see if it works across backbones or if it is just a perfect match with ChemFormer.
- In which order are the constraints removed to relax the feasible set (Sec. 2.1 - Molecule retriever)? It would be interesting to know the impact of using different ordering strategies.
- In (Sec. 2.1 - Information fusion), I see that the set of retrieved exemplars is a matrix $\in \mathbb{R}^{(\sum_{k=1}^K L_k) \times D}$, which means that you concatenate all the embeddings of the retrieved molecules. Considering that they might have different dimensions, how do you handle the mismatch? Padding? Also, doesn't this introduce an unnecessary dependence on the order in which the molecules are retrieved (which could have been avoided, e.g. by a permutation invariant aggregation)?
- I understand the model is very efficient to train. But what about preprocessing (i.e. computing all the embeddings and the pairwise similarities)? What is its impact?
- This is for my understanding: do you confirm that the iterative refinement is applied only at inference, but not during training?
- Due to the large standard deviation in Table 1b, I think it's better to also report the mode of the penalized logP values in addition to the mean.
- How come uniqueness is sometimes not reported in the generation experiments (e.g. Table 2, Table 4-right)?


**EDIT**

After a successful rebuttal, I am raising my score to 8, since the method is sufficiently new, has good experimental results, and has wide applicability across different molecular generators.

---

> ### Author Response · Authors · 2022-11-18
> **Response to Reviewer 3hQB (Part 3)**
>
> > Due to the large standard deviation in Table 1b, I think it's better to also report the mode of the penalized logP values in addition to the mean.
>
> Thanks for the suggestion! We also computed the mode of the penalized logP values for our method (i.e., the x-axis value of the largest peak in Figure A.1, Right), and we got 3.804 for $\sigma=0.6$ and 6.389 for $\sigma=0.4$. We can see our mode of the penalized logP values is still better than the mean in most of the baselines. For QMO, their mode looks very similar to ours from Figure 8 in their paper (https://arxiv.org/abs/2011.01921). We added the above discussion in Appendix C.1.
>
> > How come uniqueness is sometimes not reported in the generation experiments (e.g. Table 2, Table 4-right)?
>
> We thank the reviewer for pointing this out. The “uniqueness” metric in Table 4-left actually has the similar meaning as the “diversity” metric in Table 2 and Table 4-right. In the experiment of optimizing GSK3$\beta$ and JNK3 inhibition under QED and SA constraints (i.e., Table 2 and Table 4-right), we exactly followed the settings in [3] for a comparison, where they adopted the “diversity” terminology. For clarity, we have also provided the details of computing the “uniqueness” score in Appendix B.5.
>
> [3] Jin et al., Multi-Objective Molecule Generation using Interpretable Substructures, ICML 2020.
>
>
> -------------------------
>
> *We hope that our responses address your concerns. If we have successfully addressed your questions, we would really appreciate it if you consider raising your score. If you have any further concerns, we are happy to discuss and address them.*

---

> > ### Comment · Reviewer_3hQB · 2022-11-19
> > **Thanks for the response!**
> >
> > Thanks for the replies! Currently, I still believe this paper is borderline, although the effort you put into the rebuttal is making me lean toward complete acceptance. I feel like having some results as regards the integration with other generative methods would give this paper the breadth that is needed to be accepted, and I'll be waiting for any additional results in this sense (also I'll be looking at how you address the concerns of the other reviewers). As for your suggestion request, it is really not important which one as long as you are able to demonstrate that the approach can be applied with success also to a graph-based backbone; maybe you can try the JTVAE?
> >
> > As regards the poor exploitation of chemical space, perhaps I am misunderstanding, and if so I apologize. My belief is due to how the model is trained. Since it is trained to reconstruct the molecule nearest to the input, it appears to me that it is hardly able to explore regions of molecule space that are not represented in the training set, but just being able to cleverly interpolate between them. But let's discuss a bit to see if we can reach common ground:
> >
> >     For example, with a less strict similarity constraint, our method can generate molecules with much higher penalized logP values (see Table 1b and Figure A1, right), implying that our method can generate highly dissimilar molecules
> >
> > I am not convinced that higher penalized logP implies high dissimilarity. In fact, the most simple policy to improve logP is to add as many carbon atoms as possible, hence I can improve the logP of whatever molecule is generated by adding carbon atoms, while having the least possible dissimilarity (see https://www.ncbi.nlm.nih.gov/pmc/articles/PMC6656766/pdf/41598_2019_Article_47148.pdf)
> >
> >     In section 3.2, we show that RetMol generates more diverse molecules (Table 2) that differ from the input and from each other.
> >
> > On this I agree in principle. But to be clear, can you please explicitly state how do you compute diversity?
> >
> >     The final experiment (inhibitor design for SARS-CoV-2) also demonstrates that the optimized inhibitor in RetMol has more polar contacts and more disparate binding modes with the original compound than the Graph GA optimized inhibitor (Figure 3).
> >
> > This is a bit tricky. It is just a single example (although very impressive), I wouldn't say that it implies diversity in a so broad sense.

---

> > > ### Author Response · Authors · 2022-11-19
> > > **Response to Reviewer 3hQB's feedback**
> > >
> > > We really appreciate your response. Also, thank you for requesting further clarification on two valuable questions. We are happy to address them below.
> > >
> > > > Additional results on applying RetMol to other generative methods
> > >
> > > We thank you for your suggestion on adding these new results to improve the breadth of our work. We are working on applying RetMol to other molecule generative models and will post our results here soon. We thank you for your patience.
> > >
> > > > Further clarification on the poor exploitation of chemical space
> > >
> > > We thank you for explaining why you have concern on this point. We provide our clarification in the following:
> > >
> > > - Regarding your intuition on how the model is trained, we fully agree that our self-supervised training objective that reconstructs the nearest molecule does not necessarily encourage the exploitation of latent space. Instead, it is the **random perturbation of fused embeddings** (i.e., adding Guassian noise to fused embeddings) during inference that allows our generation to traverse in the neighborhood around the training data points (their latent embeddings, to be more precise). We do this local perturbation in every iteration and thus can ultimately result in a large-range exploitation of the chemistry space.
> > > - Regarding your comment that higher penalized logP does not imply high dissimilarity, we fully agree with it and thank you for your correction. The main motivation of using this example is to highlight that when the similarity constraint becomes weaker (i.e., $\sigma$ changes from 0.6 to 0.4), our method indeed better exploits the chemical space to be able to generate molecules with higher attribute values.
> > > - Regarding your question how `diversity` is computed, we exactly follow [1]: ${\rm diversity} = 1 - \frac{2}{n(n-1)}\sum_{X,Y} {\rm sim}(X,Y)$ where $X,Y$ are the positive generated molecules (i.e., those that satisfy the desired design criteria). That is, `diversity` measures how generated samples are different from each other. Moreover, the `novelty` metric may better reflect the “exploitation of chemical space”, which is defined as ${\rm novelty} = \frac{1}{n}\sum_X [{\rm sim}(X, X_{\rm NN}) < 0.4]$ where $X$ is a generated molecule that satisfies all the constraints and $X_{\rm NN}$ is its nearest neighbor (by Tanimoto similarity) in the training set. That is, `novelty` measures how generated samples are different from the training set. From Table 2, we see our method outperforms all previous methods regarding both `diversity` and `novelty`. We think this is strong evidence to support that our method can better exploit chemical space.
> > > - Regarding your concern on the qualitative example in the SARS-CoV-2 experiment, we agree that a single example does not imply high diversity in general. The main purpose of mentioning this example is to highlight that our method is capable of optimizing compounds beyond local edits to the scaffold, which implies our method can sufficiently exploit chemistry space, at least in the cases that we examined in this paper.
> > >
> > > We hope that our new responses address your concern. Please let us know if you have any further questions.
> > >
> > > [1] Jin et al., Multi-Objective Molecule Generation using Interpretable Substructures. ICML 2020.

---

> > > > ### Comment · Reviewer_3hQB · 2022-11-19
> > > > **Thank you again**
> > > >
> > > > Thanks for the quick and exhaustive response! Now everything is clear. Will be waiting for the additional results about the applicability of your method to graph-based molecular generators in order to improve my score further. Good luck with the experiments and thanks again for the massive effort!

---

> > > > > ### Author Response · Authors · 2022-12-05
> > > > > **Sharing our latest experimental results**
> > > > >
> > > > > We apologize for the delayed response, and thank the reviewer for the patience. Since we are approaching the end of the discussion stage, we would like to share our current experimental results about “the applicability of our method to graph-based molecular generators”.
> > > > >
> > > > > Specifically, we focused on applying our method to the hierarchical graph-to-graph translation model (HierG2G) proposed in [1], which is a more advanced graph-based molecular generator than JT-VAE (suggested by the reviewer). For illustration purpose, we considered the penalized logP optimization task under the similarity constraints, i.e. $\sigma=\{0.6, 0.4\}$, respectively (see Sec 3.1 in our paper). The average penalized logP improvement results of our method and the original HierG2G are shown as follows, where we also include JT-VAE as a reference.
> > > > >
> > > > > | Method  | $\sigma=0.6$ | $\sigma=0.4$ |
> > > > > |------------------|:--------------:|:--------------:|
> > > > > | JT-VAE  |  0.28+-0.79    |    1.03+-1.39    |
> > > > > | HierG2G |    **2.49+-1.09**   |    3.98+-1.46   |
> > > > > | RetMol (w/ HierG2G)  |     2.01+-3.90    |    **6.15+-6.73**    |
> > > > >
> > > > > We observe that 1) when $\sigma=0.4$, our method largely outperforms HierG2G, and 2) when $\sigma=0.6$, our method is slightly worse than HierG2G (but still much better than JT-VAE). Thus, although there is still a large room for improvement (see our explanation below), we believe the current results already demonstrate the applicability of our method to graph-based molecular generators.
> > > > >
> > > > > Here we explain “why our performance gets worse than HierG2G with a stricter similarity constraint”. We found that the training of the information fusion module has not converged well: the average similarity between the generated molecules and input molecules after training is only 0.39, far less than the target similarity 0.66 (see Figure A6-1). As a result, only 60% generated molecules satisfy the similarity constraint $\sigma=0.6$ (which also explains why there is a large variance in the above table). We think the insufficient training is mainly because 1) due to the limited time, we only used the default hyperparameters from HierG2G for training, which may need further optimization, and 2) HierG2G has a more complex latent space (with motif, attachment, graph substructures) than Chemformer, and thus where to apply the information fusion seems critical, but we have not explored different design choices. Therefore, we believe that fixing the training issue will further improve our results with the HierG2G backbone.
> > > > >
> > > > > We are actively continuing working on improving the training. We will incorporate the final experimental results into the revised paper in the next round.
> > > > >
> > > > > [1] Jin et al., Hierarchical Generation of Molecular Graphs using Structural Motifs, ICML 2020.

---

> > > > > > ### Comment · Reviewer_3hQB · 2022-12-07
> > > > > > **Thanks!**
> > > > > >
> > > > > > Sorry for responding just now, I didn't receive any notification (my guess, because you edited an old message). That is enough for me, your experiments show the applicability of the method with sufficiently good results. I will raise my score to 8, but I encourage you to add these results to the final version of the paper once they are finalized. Good luck!

---

> > > > > > > ### Author Response · Authors · 2022-12-08
> > > > > > > **Thank you**
> > > > > > >
> > > > > > > We thank the reviewer for recognizing the effectiveness of our new results and increasing the score! As promised, we will add the finalized version of these results to the paper.

---

> ### Author Response · Authors · 2022-11-18
> **Response to Reviewer 3hQB (Part 2)**
>
> > In section 2, you acknowledge the existence of weighting coefficients $w_l$. However, they don't seem to be present in the rest of the work. What are they (or, are they just construction to explain the problem)?
>
> We thank the reviewer for pointing this out! The weights enable a user to flexibly define their design criteria for the generative process. In all the experiments, we simply set all the weighting coefficients $w_l$ to 1, i.e., the aggregated design criterion is the sum of individual property constraints. We have added this missing information to Appendix B.2.
>
> > In which order are the constraints removed to relax the feasible set (Sec. 2.1 - Molecule retriever)? It would be interesting to know the impact of using different ordering strategies.
>
> Thanks for pointing this out! Our strategy is to relax the harder-to-satisfy constraints first and then the easier-to-satisfy ones. In other words, we first focus more on the simple attributes and then on the hard attributes (e.g., similarity -> QED -> docking), which gradually increases the level of inference difficulty over iterations. We agree that the order of the constraints being removed/relaxed is an important consideration during inference, especially when there are many of them. As suggested by the reviewer, we performed a sanity check on the impact of attribute relaxation ordering in the QED task in Sec 3.1, where, for a quick experiment, we used a subset of 200 molecules and 20 iterations. We see that if we first relax QED and then similarity, we get 89.5% success rate, and if we first relax similarity and then QED, we get 78.5% success rate. It confirms our intuition that it is preferable to relax the harder-to-satisfy constraints first and then the easier-to-satisfy ones. We added the above discussion to Appendix C.6.
>
> > In (Sec. 2.1 - Information fusion), I see that the set of retrieved exemplars is a matrix $\in \mathbb{R}^{(\sum_{k=1}^K L_k) \times D}$, which means that you concatenate all the embeddings of the retrieved molecules. Considering that they might have different dimensions, how do you handle the mismatch? Padding? Also, doesn't this introduce an unnecessary dependence on the order in which the molecules are retrieved (which could have been avoided, e.g. by a permutation invariant aggregation)?
>
> We first clarify that the different dimensions of the retrieved molecules are not a problem. Each retrieved molecule is first mapped to the embedding space with a shape of $L_k \times D$, where $L_k$ is the sequence length (whose value varies across different retrieved molecules) and $D$ is the size of the embedding vector (whose value remains the same for all retrieved molecules). Since we concatenate retrieved molecules along the sequence length dimension, we end up with a matrix with the shape of $(\sum_{k=1}^K L_k) \times D$ without padding.
>
> We then clarify that the order of retrieved molecules in aggregation is also not a problem. This is because the cross attention mechanism in the information fusion module is indeed a permutation invariant aggregation.  On a high level, the attention mechanism computes a slice of the Query embedding (shape $D$) and the encoded Key/Value matrix to obtain one slice (shape $D$) of the output matrix after inner product, softmax, and weighted sum, all of which are permutation invariant. Please see Appendix A (the information fusion part) for more details.
>
> > I understand the model is very efficient to train. But what about preprocessing (i.e. computing all the embeddings and the pairwise similarities)? What is its impact ?
>
> Preprocessing of training data is also quite efficient. Computing each embedding only involves one function evaluation of the pre-trained encoder. As for computing the pairwise similarities, we employ the highly efficient approximate kNN algorithms (e.g., SCaNN [2]), making its overhead also negligible compared to the training.
>
> [2] https://github.com/google-research/google-research/tree/master/scann.
>
> > This is for my understanding: do you confirm that the iterative refinement is applied only at inference, but not during training?
>
> This is correct. Iterative refinement would be too costly during training. Also, since the purpose of training is to encourage the information fusion module to properly integrate information from the retrieved molecules, iterative refinement is also not necessary.

---

> ### Author Response · Authors · 2022-11-18
> **Response to Reviewer 3hQB (Part 1)**
>
> We appreciate the reviewer's thoughtful comments and suggestions. We addressed the reviewer's concerns below and improved our manuscript.
>
> > Not sure it would be applicable in cases where no exemplar molecules to be retrieved are unavailable (as stated in Conclusions); this might limit its widespread applicability.
>
> This is a good point! We clarify the wide applicability of RetMol in two aspects: 1) In cases where no exemplar (positive) molecules are available, all previous methods will also be not applicable. This is because they either require a large set of positive molecules for fine-tuning or rely on a property predictor for iteration guidance, both of which are more expensive to obtain than a handful of exemplar molecules. 2) Our method has the potential to be applied to some cases where the retrieval database contains no positive molecules. For example, in the multi-property optimization tasks (Table 4, right), we show that when the retrieval database only contains molecules that satisfy the GSK3$\beta$ and JNK3 constraints but do not satisfy the QED and SA constraints, our method still achieves competitive performance (e.g., it outperforms RationaleRL). We think it is mainly because the retrieval database is continuously updated to include those that improve upon the input, which enables RetMol to generalize and extrapolate beyond the existing molecules.
>
> > It poorly exploits the chemical space, i.e. it will most likely generate molecules that are very similar to the input molecule.
>
> We respectfully disagree that our method poorly exploits the chemical space. There is much experimental evidence. For example, with a less strict similarity constraint, our method can generate molecules with much higher penalized logP values (see Table 1b and Figure A1, right), implying that our method can generate highly dissimilar molecules. In section 3.2, we show that RetMol generates more diverse molecules (Table 2) that differ from the input and from each other. The final experiment (inhibitor design for SARS-CoV-2) also demonstrates that the optimized inhibitor in RetMol has more polar contacts and more disparate binding modes with the original compound than the Graph GA optimized inhibitor (Figure 3).
>
> > What is the justification for having used ChemFormer instead of any other encoder-decoder architecture as the backbone model? It would be great to see RetMol applied to another backbone, to see if it works across backbones or if it is just a perfect match with ChemFormer.
>
> This is a good point! Since the main focus of this work is to demonstrate the effectiveness of a retrieval-based approach for controllable molecule generation, we did not systematically investigate the impact of various generative model backbones on the performance. The main reason for using ChemFormer in this work is to highlight that our method can work well with varying-dimensional latent-variable generative models, where many previous works (such as a class of latent optimization-based methods) that require a fixed-dimension latent space are not applicable. As transformer-based large language models (LLMs) are becoming increasingly popular for modeling molecules, we think it is important to demonstrate the compatibility of RetMol to LLMs, such as ChemFormer.
>
> On the other hand, we thank the reviewer for the valuable suggestion. We are actively working on extending our methodology to additional molecule generative models, such as those based on molecule graph structures [1]. Though the extension is straightforward on a high level, we have to admit that there is much engineering effort involved, mainly because 1) different backbone models (especially those that leverage the graph structures) process input data differently so we need to adapt our data preparation and processing accordingly, and 2) different environments and package versioning cause inconsistencies, (e.g., in some cases rdkit needs to be a specific earlier version to work: see [issue 1](https://github.com/wengong-jin/hgraph2graph/issues/44), [issue 2](https://github.com/wengong-jin/hgraph2graph/issues/18), [issue 3](https://github.com/wengong-jin/hgraph2graph/issues/20)). We are currently solving these engineering challenges and will share any new results in time. Meanwhile, if you have any suggestion on the molecule generative models to try, please let us know.
>
> [1] Jin et al., Hierarchical Generation of Molecular Graphs using Structural Motifs,ICML 2020.

---

### Official Review · Reviewer_PZ6g · 2022-10-24

**Confidence:** 3
**Correctness:** 3
**Technical Novelty And Significance:** 2
**Empirical Novelty And Significance:** 2
**Recommendation:** 6

**Clarity, Quality, Novelty And Reproducibility:**

# Clarity
This paper is well-written and easy to follow

# Novelty
The authors fail to provide a comprehensive literature survey on general retrieval-based training methods, especially those in NLP and CV. For example, in “Instance-Conditioned GAN, NeurIPS 2021”, the authors show that a retriever can improve the generation capabilities with very few data points. The authors claim briefly the way the retrieval modules retrieve and integrate the information is slightly different from MSA, but it is not clear to me where are those differences: The retrieval pipeline looks quite standard and the fusion module is just based on the standard attention mechanism. It is true that the authors in this submission focus on applications to bio-related tasks, but the methods used here are not modified accordingly, and thus I think there are very limited technical contributions.

# Reproducibility
The authors do not provide the source code in the supplement., and they do not mention in the paper if they will release their code, making it difficult for others to reproduce this work.

**Strength And Weaknesses:**

# Strength

- The experiments are very extensive with lots of details, including both qualitative and quantitative materials.
- It evaluates the effectiveness of simple retrieval modules for controllable molecule generation.

# Weakness


- Inference via iterative refinement introduced in Section 2.3 is quite heuristic, yet the heuristic is not designed carefully. The authors mention that iterative update is common in approaches such as GA methods, and there are many off-the-shelf GA tools. It seems to me that it’s a better choice to use those tools instead of relying on a design with much simpler heuristics.

- On most benchmark tasks, RetMol does not significantly outperform other baselines, and those baselines methods are usually proposed in 2019. The authors should include recent stronger baselines to justify the value of retrieval modules.

**Summary Of The Paper:**

Motivated by the need for sample-efficient molecule generation, this paper introduces a simple retrieval mechanism to retrieve and fuse the exemplar molecules for controllable molecule generation. Evaluated on a variety of synthetic molecule generation tasks, the proposed method outperforms other baselines.

**Summary Of The Review:**

In summary, this paper proposes a retrieval-based method for controllable molecule generation and extensively evaluates its effectiveness on benchmark datasets. The introduction of a retriever for molecule generation is new as far as I know, but it’s not new in other ML communities. However, the design of the retrieval components does not bring in any technical contributions to the community and is mostly based on ungrounded heuristics.

---

> ### Author Response · Authors · 2022-11-18
> **Response to Reviewer PZ6g (Part 2)**
>
> > The authors fail to provide a comprehensive literature survey on general retrieval-based training methods, especially those in NLP and CV.  (IC-GAN, MSA Transformer)
>
> We thank the reviewer for suggesting IC-GAN as related work. To provide a more comprehensive literature survey, we have added IC-GAN and some more recent related works in NLP and CV to Sec 4, including retrieval-augmented diffusion models [1,2] and retrieval-augmented language models [3,4]. We have also more clearly discussed the major difference between RetMol and these methods.
>
> Regarding the key difference between RetMol and the MSA, the MSA methods (e.g., MSA Transformer) focus on the pairwise interactions among a set of evolutionarily related (protein) sequences while RetMol considers the cross attention between the input and a set of retrieved examples. We have added this discussion to Sec 4.
>
> [1] Qi et al., Semi-parametric Image Synthesis, Apr 2022.
> [2] Chen et al., Re-Imagen: Retrieval-Augmented Text-to-Image Generator, Sep 2022.
> [3] Wu et al., Memorizing Transformers, Jun 2022.
> [4] Zhang et al., GreaseLM: Graph REASoning Enhanced Language Models for Question Answering, Jan 2022.
>
> > It is true that the authors in this submission focus on applications to bio-related tasks, but the methods used here are not modified accordingly, and thus I think there are very limited technical contributions.
>
> We respectfully disagree with the reviewer that there are very limited technical contributions in our work. Achieving controllable generation with the retrieval-based approach is an emerging and underexplored direction even in the NLP and vision domains. Although there exist many prior works on retrieval-based generative models, the majority of them are not for controllable generation. Instead, the retrieval mechanism among them is mainly used for improving the generation quality either with a smaller model or with very few data points. It is non-trivial to extend retrieval-based methods for controllable generation, as it requires a development of novel training and inference algorithms for the retrieval module (as we did in Sec 2.2 and Sec 2.3).
>
> Our work is among the first few that successfully demonstrates the utility and promise of retrieval-based methods in the context of (multi-attribute based) controllable generation, and is the first one for controllable molecule generation. Indeed, the design of our retrieval components (e.g., retriever and information fusion) follows from the standard design in previous works. On top of this, our main technical contributions include 1) a new retrieval-based controllable generation framework that requires training on the information fusion module only without modifying the rest of the model parameters, 2) a new self-supervised training method (as opposed to the classical self-reconstruction objective) that generalizes to various downstream tasks without any re-training; and 3) a new iterative refinement approach that not only refines the generated samples, but also updates the retrieval database for better generalization.
>
> While there is plenty of room for future research, we believe that our work brings novel ideas and technical contributions to research on retrieval-based controllable generation.
>
> -------------------------
> *We hope that our responses address your concerns. If we have successfully addressed your questions, we would really appreciate it if you consider raising your score. If you have any further concerns, we are happy to discuss and address them.*

---

> > ### Comment · Reviewer_PZ6g · 2022-11-19
> > **Keep my ranting unchanged**
> >
> > Thanks for the response!
> >
> > I agree that the application of retrieval-based methods seems novel in the context of controllable molecule generation. However, I still think the technical contributions are limited, and the retrieval module is not modified according to molecule generation tasks in my opinion. Therefore, I will keep my rating unchanged.

---

> > > ### Author Response · Authors · 2022-11-19
> > > **Response to Reviewer PZ6g’s feedback**
> > >
> > > We thank you for your response. We appreciate that the reviewer agrees with the novelty in the application of retrieval-based methods to controllable molecule generation. We would like to share more our thoughts on the technical novelty/contribution of our work:
> > > - Technically, it is non-trivial to apply retrieval-based methods to controllable generation. To make retrieval mechanism really applicable to various controllable generation tasks, we propose 1) **a novel self-supervised method** that predicting most similar molecule based on the input molecule and other K−1 similar molecules from the retrieval database, which generalizes well to various downstream tasks *without retraining*, and 2) **a novel iterative inference process** that dynamically updates the retrieval database with the generated molecules, which enables us to *extrapolate beyond the best of the dataset*. **To the best of our knowledge, none of these two technical contributions appears in previous retrieval-based methods.**
> > > - Regarding the reviewer’s concern that our method does not modify the retrieval module, we believe **simplicity is difficult and important**, in particular when our method largely outperforms previous methods. Empirically, we also experimented with the more advanced information fusion strategy, i.e., applying `the row and column attention` between the input molecule and retrieved molecules (inspired by the MSA transformer work [1]; see their Figure 1 for more details), insead of the standard cross-attention in Eq. (1) (which can be viewed as applying the row attention only). However, we did not observe any significant improvements so we still adopted the standard across attention strategy in our method. We strive to make our method as simple as possible to really highlight what the most important design choices are when applying retrieval to controllable (molecule) generation.
> > >
> > > [1] Rao et al., MSA Transformer, ICML 2021.

---

> > > > ### Comment · Reviewer_PZ6g · 2022-11-20
> > > > **Not enough**
> > > >
> > > > I understand that "novelty" is sometimes overused in a review process, and I'm aware of this. I agree that simplicity is important and novelty does not necessarily mean complicated methods. However, retrieval-based methods are extensively studied in the NLP community and I feel that the authors are not aware of them. For example, the authors mention "a novel self-supervised method that predicting most similar molecule based on the input molecule and other K−1 similar molecules from the retrieval database, which generalizes well to various downstream tasks without retraining". This is similar to [1], in which they both use the current input and the retrieved data to perform some tasks.
> > > >
> > > > As for the iterative inference process, this seems new for retrieval-based generation tasks. But it is a common practice for biological sequence design, e.g., [2], where the generated sequences will be added to the database and re-train the model. This process is similar to the proposed one, and the difference is in this submission the new data is used for retrieval.
> > > >
> > > >
> > > > [1] GNN-LM: Language Modeling based on Global Contexts via GNN, ICLR 2022
> > > >
> > > > [2] Model-based reinforcement learning for biological sequence design, ICLR 2020

---

> > > > > ### Author Response · Authors · 2022-11-20
> > > > > **Thank you for your prompt response**
> > > > >
> > > > > We thank the reviewer for providing more context to make us better understand your concern about technical novelty.
> > > > >
> > > > > > However, retrieval-based methods are extensively studied in the NLP community and I feel that the authors are not aware of them. For example, the authors mention "a novel self-supervised method …". This is similar to [1], in which they both use the current input and the retrieved data to perform some tasks.
> > > > >
> > > > > Regarding “retrieval-based methods are extensively studied in the NLP community”, we fully agree with it. We have cited several representative works in the Related Work section, such as language modeling (Borgeaud et al., 2021; Liu et al., 2022; Wu et al., 2022), code generation (Hayati et al., 2018), question answering (Guu et al., 2020; Zhang et al., 2022), etc., and discussed our main differences with them. We are happy to cite more if the reviewer thinks we missed other important references in the NLP community.
> > > > >
> > > > > Regarding our similarity to [1], we believe any retrieval-based method will “use the current input and the retrieved data to perform some tasks”. In particular, [1] used the original input tokens and the retrieved similar tokens to build a heterogeneous graph, which aggregates information from similar contexts to improve the language modeling performance. Therefore, [1] is similar to us regarding the retrieval criterion *during training*: given an input, we both used the *similarity* score to retrieve K most similar samples. However, there are key differences:
> > > > > - The problem under investigation is different, even though retrieval methods are involved as part of the technical approach. Like other retrieval-based methods in NLP that we mentioned in the paper (see the Related Work section), [1] studies the problem of improving the generation quality of vanilla LMs. In contrast, we studied the problem of controllable generation.
> > > > > - Due to the difference of underlying tasks, the training objective is different. Specifically, with K retrieved samples, [1] still applies the conventional reconstruction objective (i.e., the standard language modeling) during training. However, our training objective is to predict the nearest neighbor (i.e., the most similar sequence in the retrieval database) of the input. In our setting, reconstructing the input sequence will not be able to update the information fusion. Please see the “Remarks” paragraph in Section 2.2 for more insights into our design choice.
> > > > >
> > > > > > But it is a common practice for biological sequence design, e.g., [2], where the generated sequences will be added to the database and re-train the model. This process is similar to the proposed one, and the difference is in this submission the new data is used for retrieval.
> > > > >
> > > > > We believe that the synergy between the retrieval mechanism and the iterative refinement process (to perform controllable generation) is novel to both the retrieval-based generation community and the molecule optimization community. In particular, our idea of dynamically updating the retrieval database (i.e., “the new data is used for retrieval”) over iterations is the key to the success of this synergy in various molecule optimization tasks.
> > > > >
> > > > > We agree that the multiple rounds of model-based RL training in [2] shares the similarity with our inference design: 1) both are iterative, and 2) both use previously generated data in the next iteration. However, there are also key differences:
> > > > > - *(Setting)*. [2] employs the iterative process during training while our work does it during inference. In particular, we do not update any parameters during inference. When adapting the method to different tasks, [2] requires *intensive* re-training while we only need to update the retrieval database and perform the similar inference process without re-training.
> > > > > - *(Methodology)*. Since [2] is not a retrieval-based method, it does not use the generated data to augment the retrieval database. Also, in each round, our method could retrieve from the set of *all the best-generated data in the iteration history* for better inference. Instead, in each round, [2] only uses *the best-generated data from the last round* to update model parameters.
> > > > > - *(Task)*. [2] focuses on DNA and protein sequence design while we focus on small molecule design. The main challenge in [2] is to deal with the large search space for optimization, while our main challenge is to satisfy multiple property constraints. Please see the Related Work section in [2] for more discussions.
> > > > >
> > > > > In summary, we believe although [1,2] share some similarity with our method from different aspects, there exist key differences as discussed above. We are happy to incorporate the above discussions to the revised paper in the next round.
> > > > >
> > > > > We hope our response further clarifies the reviewer’s concerns regarding the technical novelty. If you have any further questions, we are more than happy to answer them.

---

> > > > > > ### Comment · Reviewer_PZ6g · 2022-11-30
> > > > > > **Increasing score to 6**
> > > > > >
> > > > > > Thanks for your responses. After reading other reviewers' comments and AC's suggestions, I agree that emphasizing too much on novelty seems a bit too harsh. I decide to increase the score to 6. But I still suggest discussing [1,2] in your final version.

---

> > > > > > > ### Author Response · Authors · 2022-12-01
> > > > > > > **Thank you**
> > > > > > >
> > > > > > > We thank the reviewer for increasing the score! We also thank the reviewer for the suggestion and will incorporate the above discussions with [1,2] into the final version.

---

> ### Author Response · Authors · 2022-11-18
> **Response to Reviewer PZ6g (Part 1)**
>
> We appreciate the reviewer's thoughtful comments and suggestions. We addressed the reviewer's concerns below and improved our manuscript.
>
> > Inference via iterative refinement introduced in Section 2.3 is quite heuristic, yet the heuristic is not designed carefully. The authors mention that iterative update is common in approaches such as GA methods, and there are many off-the-shelf GA tools. It seems to me that it’s a better choice to use those tools instead of relying on a design with much simpler heuristics.
>
> We would like to point out that our iterative refinement process, on a high-level, shares a similar design principle to that of GA-based methods and other prior works (i.e., input → update → generate → select → input → …). Our main differences with GA lie in the “update” step: 1) we need to incorporate retrieved information in the process through the information fusion module, and 2) we also need to update the retrieval database over iterations. Due to the above two important features in our method, an off-the-shelf GA tool cannot be applied directly to RetMol.
>
> Specifically, GA directly perturbs the input molecules via heuristics including mutations and crossovers in the “update” step. Our approach uses a learnable information fusion module to fuse the input molecules and retrieved molecules, and then adds a standard Gaussian noise to the fused embedding in the “update” step. As we can see, our design instead removes the need of hand-crafted heuristics in GA (e.g., mutations and crossovers) in the inference process. Besides, GA does not update the retrieval database but we do, which makes our method better generalize beyond the best of dataset. Therefore, our iterative refinement process is compatible with the retrieval mechanism, and is more advanced and less heuristic than GA.
>
> > On most benchmark tasks, RetMol does not significantly outperform other baselines, and those baselines methods are usually proposed in 2019. The authors should include recent stronger baselines to justify the value of retrieval modules.
>
> We respectfully disagree that the baseline methods are usually proposed in 2019 and are not strong. We revisited each task in the following:
>
> - In Table 1 (Sec 3.1), we compared with QMO (2021). Our method consistently outperformed QMO (2021) across different settings, and in particular achieved up to 49.8% average penalized logP improvement.
> - In Table 2 (Sec 3.2), we compared with rationaleRL (2020) and MARS (2021). We also added a new baseline comparison (MolEvol, 2021) per reviewer 9XRZ’s suggestion. Our method consistently outperformed all these baselines for all three metrics (success rate, novelty and diversity), and in particular improved the success rate over the best method MARS (2021) by 4.6% (96.9% vs. 92.3%).
> - In Figure 2 (Sec 3.3), we note that on the Guacamol benchmark, the baselines we compared with are the best ones so far in the leaderboard (https://www.benevolent.com/guacamol, GOAL DIRECTED BENCHMARKS), even though they were published 3 years ago. For example, the latest work (https://arxiv.org/abs/2210.16099) reported a score of 0.654 and 0.491 for perindopril MPO and zaleplon MPO, respectively (Table D1 in their paper), which is on par with SMILES GA (0.661 and 0.413, respectively), and less competitive against Graph GA (0.792 and 0.754, respectively) and our proposed RetMol (0.765 and 0.713, respectively).
> - In Table 3 (Sec 3.4), we compared with Graph GA (2019) and QMO (2021). Note that since QMO (2021) completely failed in generating molecules that satisfy all properties, we didn’t include its results in Table 3 (which we mentioned in the second paragraph of Sec 3.4). We chose Graph GA (2019) because it is one of the strongest baselines in multi-property optimization tasks, as shown on the Guacamol benchmark. We see that compared with the best baseline, our framework succeeds more often at generating new molecules with the given design criteria (62.5% vs. 37.5% success rate) and can generate on average more potent optimized drug molecules (2.84 vs. 1.67 kcal/mol average binding affinity improvement over the original drug molecules).

---

### Official Review · Reviewer_9XRZ · 2022-10-29

**Confidence:** 3
**Correctness:** 3
**Technical Novelty And Significance:** 3
**Empirical Novelty And Significance:** 3
**Recommendation:** 6

**Clarity, Quality, Novelty And Reproducibility:**

This paper is mostly clear. The authors did not release the code but promised to release it in the future.

**Strength And Weaknesses:**

Strength:
- Data efficiency is a practical issue in the optimization of molecular properties. The proposed method could be used to optimize molecules given a small number of positive examples.
- To my knowledge, the self-supervised training strategy of predicting the nearest neighbors given the remaining neighbors is novel. The task-agnostic training makes it depend less on the number of task-specific samples.

Weaknesses:
- The paper did not justify why such self-supervised training strategy would work since the inference time objective is not aligned with the training time objective.
  - During the training time, the information fusion is trained to predict the nearest molecule given the K-1 remaining nearest neighbors. An information fusion model that adds a small noise to the current molecule would yield a high objective score, which means the information fusion model learns to predict a similar molecule to the current one.
  - However, during inference time, the model is asked to predict a similar molecule but in the direction of the positive examples. The inference time objective is quite different from the training time objective. This is a mismatch that needs further explanation and/or empirical analysis in the paper.
- The design of the iterative refinement approach is ad-hoc, and the description in Section 2.3 seems to miss important details about random perturbing.
  - From my understanding, the model is expected to interpolate between the current molecule and the positive examples in the axis of both structural similarity and property scores. It’s unrealistic to expect the model to extrapolate beyond the database, even if we do iterative refinement. The key mechanism to allow local exploration is random perturbing, mentioned briefly in Section 2.3. However, details are missing here.
- The abstract and intro section gives the readers the wrong impression that the proposed method only requires a small number of samples for a property optimization task. However, the method still requires a large number of samples (number of calls to the property prediction), even in the SARS-COV-2 experiment where there are only 23 positive examples given. Since this paper focuses on sample efficiency, it would be more convincing if the authors could provide the exact numbers of samples used in the proposed method and all baselines for a full comparison. For example, in the multi-property optimization experiment in Section 3.2, the number of samples required by each methods are (copied from the Appendix):
  - 296, 000 for the proposed method.
  - 2, 750, 000 for MARS [1].
  - 200, 000 for MolEvol [2] (missing in the baselines).
  - xxx for RationaleRL[3].

[1] Xie, Yutong, et al. "MARS: Markov Molecular Sampling for Multi-objective Drug Discovery." International Conference on Learning Representations. 2021.
[2] Chen, Binghong, et al. "Molecule optimization by explainable evolution." International Conference on Learning Representation (ICLR). 2021.
[3] Jin, Wengong, Regina Barzilay, and Tommi Jaakkola. "Multi-objective molecule generation using interpretable substructures." International conference on machine learning. PMLR, 2020.


**Summary Of The Paper:**

This paper proposed a retrieval-based method for optimizing molecular properties under similarity constraints. The proposed method leverages a pre-trained encoder-decoder generative model and trains an information fusion module to steer the pre-trained model toward generating more similar molecules to the retrieved ones. During training time, the model is asked to predict the nearest molecule in the retrieved ones. And then, during inference time, an iterative refinement process is employed to perform multi-step optimization. The proposed method performed competitively on several molecular optimization tasks using only a small set of exemplar molecules.

**Summary Of The Review:**

The paper proposed a retrieval-based method for molecular property optimization. Although the proposed method achieved competitive results, the methodology is less principled and needs more work.

---

> ### Author Response · Authors · 2022-11-18
> **Response to Reviewer 9XRZ (Part 2)**
>
>
> > The description in Section 2.3 seems to miss important details about random perturbing.
>
> We agree that the random perturbation of fused embeddings enables the generator to explore the chemical space and extrapolate beyond the database. In this work, we employ the perturbation by simply adding independent isotropic Gaussian noises with zero mean and standard deviation of 1 to each fused embedding. We originally mentioned the above detailed information about random perturbation in Appendix B.2, and to make the main text more clear, now we also added it to Sec. 2.3.
>
>
> > The abstract and intro section gives the readers the wrong impression that the proposed method only requires a small number of samples for a property optimization task. Since this paper focuses on sample efficiency, it would be more convincing if the authors could provide the exact numbers of samples used in the proposed method and all baselines for a full comparison.
>
> We believe there might be a bit of misunderstanding here regarding sample efficiency. We stated in the abstract and intro that our approach uses “a small set of exemplar molecules” from the retrieval database, which aims to highlight that one does not need a large retrieval database to achieve good controllable generation results. In other words, we emphasize that the construction of the retrieval database in our method is far less costly than many previous methods that require training/fine-tuning the pre-trained models with a large number of annotated samples for each task.
>
> Indeed, we also used the property predictor to guide the inference process, by following the same procedure of baseline methods (e.g., GA/QMO). As for the number of generated samples (or number of calls to the property prediction), we have provided the details of our method and all baselines in Appendices B.4-B.7:
>
> - In most of the experiments, we keep the number of generated samples the same for both our approach and the best baselines for a fair comparison. For example, for QED and penalized logP experiments, we used 50,000 and 80,000 samples, respectively, which are exactly the same with QMO (see Appendix B.4). For the SARS-COV-2 experiment, both Graph-GA and our method used 10,000 samples (see Appendix B.7).
> - For the experiment in Section 3.2, as the reviewer suggested, we provide the complete comparison of the competitive methods regarding the performance and the number of generated samples as follows, where we can see that our method is sample efficient with the best performance.
>
> | Method  | Success % | Novelty | Diversity | Number of generated samples |
> |------------------|:--------------:|:--------------:|:--------------:|:--------------:|
> | RationaleRL  |  74.8    |    0.568    |    0.701  |       1,086,000       |
> | MARS |    92.3    |    0.824    |    *0.719*    |   2,750,000       |
> | MolEvol   |   *93.0*    |    0.757    |    0.681    |     **200,000**      |
> | RetMol  |     **96.9**    |    **0.862**    |    **0.732**    |  *296,000*       |
>
>
> We thank the reviewer for pointing out the MolEvol work. We have revised the paper and it is now included in Table 2, where our method consistently outperforms MolEvol on all three metrics (success rate, novelty, diversity). We also updated Appendix B.5 with the complete list of the number of generated samples for each competitive method mentioned above.
>
>
> -------------------------
> *We hope that our responses address your concerns. If we have successfully addressed your questions, we would really appreciate it if you consider raising your score. If you have any further concerns, we are happy to discuss and address them.*

---

> > ### Comment · Reviewer_9XRZ · 2022-12-02
> > **Raise my score to 6**
> >
> > Thank you for the response! It clarifies most of my concerns.
> >
> > > An information fusion model that adds a small noise to the current molecule would yield a high objective score, which means the information fusion model learns to predict a similar molecule to the current one.
> >
> > Here I meant a local minimum of this self-supervised training is that the model learns to generate a molecule that is very similar to the input molecule. Because $x_{1NN}$ is the nearest neighbor, we can expect that such a prediction would yield a decent score.
> >
> > Considering the novelty and strong empirical performance, I will raise my score to 6. However, the support for the theoretical side of the approach is still weak. I suggest the authors continue improving this part to make the method more principled.

---

> > > ### Author Response · Authors · 2022-12-03
> > > **Thank you**
> > >
> > > We thank the reviewer for raising the score! Regarding the proposed self-supervised training objective, we agree that a theoretical explanation of its generalization phenomenon will make our method more principled. We will follow the reviewer's suggestion to explore more principled insights into understanding the successes and limitations of our method.

---

> ### Author Response · Authors · 2022-11-18
> **Response to Reviewer 9XRZ (Part 1)**
>
> We appreciate the reviewer's thoughtful comments and suggestions. We addressed the reviewer's concerns below and improved our manuscript.
>
> > The paper did not justify why such self-supervised training strategy would work since the inference time objective is not aligned with the training time objective.
>
> We thank the reviewer for pointing this out! In the following, we first clarify a misunderstanding of "small noise" during training and then justify why our self-supervised training works.
>
> > During the training time, the information fusion is trained to predict the nearest molecule given the K-1 remaining nearest neighbors. An information fusion model that adds a small noise to the current molecule would yield a high objective score, which means the information fusion model learns to predict a similar molecule to the current one.
>
> We would like to clarify that there is no noise involved during training, i.e., the information fusion module simply aggregates the embeddings from the input and the retrieved molecules and passes the fused embedding directly to the decoder. The "small noise" part, i.e., random perturbation of the fused embedding, only comes into picture during inference as a mechanism to encourage the generator to explore the chemical space.
>
> > However, during inference time, the model is asked to predict a similar molecule but in the direction of the positive examples. The inference time objective is quite different from the training time objective. This is a mismatch that needs further explanation and/or empirical analysis in the paper.
>
> We provide justifications for why the mismatch of objectives during training and inference is not an issue as follows:
>
> - In the whole retrieval-based controllable generation pipeline, the information fusion module is only responsible for aggregating different sources of information coherently (i.e., no matter where they come from) while it is the retriever that actually takes care of the underlying objective (i.e., controlling the sources of information). As long as the retriever (and the retrieval database) is task-specific, we believe our method can be easily applied to different tasks.
> - In most downstream tasks on molecule optimization, besides satisfying various desired molecular properties, we still want to maintain the structural similarity of the output molecule with the input molecule. That is, the similarity objective always appears as a part of various inference objectives. Thus, we think using the similarity objective during training can better generalize across downstream tasks without re-training.
> - Empirically, our extensive experiments justify that our self-supervised training strategy works well in diverse controllable generation settings. Moreover, recent progress on CV and NLP (e.g., MAE [1], BERT [2] and CLIP [3]) also suggests that a properly designed self-supervised training objective can lead to improvements for a number of downstream tasks whose inference-time objectives differ from that during training.
>
> Investigations into theoretical explanations of this generalization phenomenon are still under active research and we leave this as an important future work for retrieval-guided controllable generation.
>
> [1] He et al., Masked Autoencoders Are Scalable Vision Learners, CVPR 2022.
> [2] Devlin et al., BERT: Pre-training of Deep Bidirectional Transformers for Language Understanding, NAACL 2019.
> [3] Radford et al., Learning Transferable Visual Models From Natural Language Supervision, ICML 2021.
>
> > The design of the iterative refinement approach is ad-hoc.
>
> We respectfully disagree that our interactive refinement is ad-hoc. On a high-level, our iterative refinement process shares a similar design principle to that of GA and other prior works, such as QMO (i.e., input → update → generate → select → input → …). Our main differences with GA/QMO lies in the “update” step: 1) we need to incorporate retrieved information in the process through the information fusion module, and 2) we also need to update the retrieval database over iterations.

---

### Official Review · Reviewer_UNcN · 2022-12-07

**Confidence:** 5
**Correctness:** 4
**Technical Novelty And Significance:** 3
**Empirical Novelty And Significance:** 2
**Recommendation:** 6

**Clarity, Quality, Novelty And Reproducibility:**

- description is reasonable clear
- novelty is given but limited
- quality good enough for ICLR
- overall, results seems to be reproducible

**Strength And Weaknesses:**

strengths
- comprehensive evaluation
- new, reasonably motivated model

weaknesses
- ML novelty somewhat limited
- docking as oracle is not ideal

**Summary Of The Paper:**

The paper studies molecule generation, proposing a new model using a retrieval mechanism, which has recently shown to be successful in NLP.

The method is evaluated on several established benchmarks and reaches performance comparable to state of the art

**Summary Of The Review:**

good evaluation, decent results, limited novelty

---

> ### Author Response · Authors · 2022-12-14
> **Response to Reviewer UNcN**
>
> Thank you for the review. We appreciate the reviewer's comments that our method is "new, reasonably motivated", our results are "decent", and the work has "quality good enough for ICLR". We will add more discussions with other retrieval-based methods (particularly those in NLP) to better highlight our method's key differences.

---

### Decision · Program_Chairs · 2023-01-20

**Decision:**

Accept: notable-top-25%

**Justification For Why Not Higher Score:**

The paper may be only interesting to a particular domain.

**Justification For Why Not Lower Score:**

All reviewers appreciate the paper's contribution and discussions with the authors.

**Metareview: Summary, Strengths And Weaknesses:**

The paper proposes a retrieval-based approach for generating molecules. The method is similar to retrieval-based language modelling. The pre-trained generation model is used to predict the nearest molecules satisfying desired properties (drug-likeness etc). The retrieved molecules are then fused to iteratively update the generation. The proposed approach is evaluated in three controllable generation scenarios, one real-world drug design scenario (for SARS-COVID protease), and the GuacaMol benchmark. The method achieves improvement over a range of different metrics such as success rate or docking score.

Strengths of the paper:
1. Novel application of retrieval-based generation in molecule domain
2. Iterative refinement with retrieved nearest molecules.
3. comprehensive experiments on a variety of targets.

Weakness of the paper:
1. Justification of self-supervised training
2. the experiment shows it improves the diversity and novelty of generated molecules, which is a bit counter-intuitive to retrieval-based methods. From the intuition, the method should generate molecules that are very similar to the input molecules.
3. design of iterative refinement can be improved.

Overall, the paper has descent technical contribution, experimental support, and is well-written.

**Note From Pc:**

if the above contains the word "oral" or "spotlight" please see: "oral" presentation means -> notable-top-5% and "spotlight" means -> notable-top-25%. As stated in our emails, we are disassociating presentation type from AC recommendations

**Summary Of Ac-Reviewer Meeting:**

N/A